



# Towards a community-wide effort for benchmarking in subsurface hydrological inversion: benchmarking cases, high-fidelity reference solutions, procedure and a first comparison

Teng Xu[1], Sinan Xiao[2], Sebastian Reuschen[3], Nils Wildt[3], Harrie-Jan Hendricks Franssen[4,5], and Wolfgang Nowak[3]

[1]College of Water Conservancy and Hydropower Engineering, Hohai University, Nanjing, China
[2]Department of Mathematical Sciences, University of Bath, Claverton Down, Bath, BA2 7AY, United Kingdom
[3]Institute for Modelling Hydraulic and Environmental Systems, University of Stuttgart, 70569 Stuttgart, Germany
[4]Institute of Bio- and Geosciences: Agrosphere (IBG-3), Forschungszentrum Jülich GmbH, 52425 Jülich, Germany
[5]Centre for High Performance Scientific Computing in Terrestrial Systems: HPSC TerrSys, Geoverbund ABC/J, Leo Brandt Strasse, Jülich, Germany

**Correspondence:** Wolfgang Nowak (wolfgang.nowak@iws.uni-stuttgart.de)

**Abstract.** Inversion in subsurface hydrology refers to estimating spatial distributions of (typically hydraulic) properties, often associated with quantified uncertainty. Many methods are available, each characterized by a set of assumptions, approximations, and numerical implementations. Only a few intercomparison studies have been performed (in the remote past) amongst different approaches (e.g., Zimmerman

et al., 1998; Hendricks Franssen et al., 2009). These intercomparisons guarantee broad participation to push forward research efforts of the entire subsurface hydrological inversion community. However, in past studies until now, comparisons were made among approximate methods without firm reference solutions. Without reference solutions, one can only compare competing best estimates and their associated uncertainties in an intercomparison sense, and absolute statements on accuracy are unreachable.

Our current initiative defines benchmarking scenarios for groundwater model inversion. These are targeted for community-wide use as test cases in intercomparison scenarios. Here, we develop five synthetic,



open-source benchmarking scenarios for the inversion of hydraulic conductivity from pressure data. We also provide highly accurate reference solutions produced with massive high-performance computing and with a high-fidelity MCMC-type solution algorithm. Our high-end reference solutions are publicly

available, as well as the benchmarking scenarios, the reference algorithm, and suggested benchmarking metrics. Thus, in comparison studies, one can test against high-fidelity reference solutions rather than discussing different approximations.

To demonstrate how to use these benchmarking scenarios, reference solutions, and suggested metrics, we provide a blueprint comparison of a specific ensemble Kalman filter version. We invite the community

to use our benchmarking scenarios and reference solutions now and into the far future in a community-wide effort towards clean and conclusive benchmarking. For now, we aim at an article collection in an appropriate journal, where such clean comparison studies can be submitted together with an editorial summary that provides an overview.

## 1 Introduction

Subsurface flow processes take place in geologic media where heterogeneities occur across a multiplicity of scales. These cause the values of relevant hydrogeological parameters to vary over several orders of magnitude. When considering this heterogeneity together with the fact that we have a limited ability to look deep into the subsurface, it is easy to understand the reasons why predicting the behavior of subsurface environments is fraught with remarkable uncertainty.



Only data can assist in reducing this uncertainty. In practice, direct measurements of model parameters
(such as hydraulic conductivity, porosity, or storativity) are scarce and only available at a limited number
of locations. Additionally, such data are defined or measured as scales inconsistent with the used model,
and they are subject to measurement inaccuracies and imprecision. Hence, one must employ indirect
information, i.e., observations of state variables such as pressure heads and/or concentrations. To embed
these data in a procedure to estimate system parameters is typically termed model calibration or inverse
modeling. Solving the inverse problem is at the heart of characterizing complex natural systems.

Data scarcity and subsurface heterogeneity are responsible for the uncertainty that remains even after
calibration. The latter uncertainty is critical and must be quantified to provide robust decision support
in engineering and management practice. Stochastic (or probabilistic) inverse approaches (also known as
conditional simulation, Bayesian updating, and statistical parameter inference) are the only way to obtain
such uncertainty estimates. Providing algorithms for solving stochastic inverse problems acceptably fast,
robustly, and accurately is one of the remaining key challenges in modern subsurface characterization
approaches. In recent years, a remarkable variety of inverse modeling approaches has been proposed.
The most popular approaches, such as geostatistical regularization (de Marsily, 1978), pilot points and
sequential self-calibration (e.g., RamaRao et al., 1995; Gómez-Hernández et al., 1997, 2001; Hendricks
Franssen et al., 1999, 2003), quasi-linear geostatistical approaches (e.g., Kitanidis, 1995; Nowak and
Cirpka, 2006; Schwede and Cirpka, 2009), MCMC-based approaches (e.g., Laloy et al., 2013; Cotter
et al., 2013b; Xu et al., 2020), ensemble Kalman filters (EnKFs) and ensemble smoothers (e.g., Evensen,


2003; Van Leeuwen and Evensen, 1996; Xu et al., 2021), have been applied in many fields. However, a

conclusive and convincing assessment of their relative merits and drawbacks is still lacking, especially in

the area of groundwater modeling. This is mostly related to the fact that there are no well-defined bench-

marking scenarios for rigorous comparison under standardized, controlled, and reproducible conditions.

There are only two large comparison studies, in which many (i.e. seven in both studies) methods were

compared against each other (Zimmerman et al., 1998; Hendricks Franssen et al., 2009). Although very

useful, these two comparison studies still suffer from several limitations: 1) several more recent methods

were not included (e.g., all EnKF-based methods and non-multi-Gaussian inversion approaches); 2) the

candidate solutions were not compared against reference inverse solutions (obtained, e.g., with brute-force

Monte Carlo approaches), but to synthetic data-generating realities; this also implies that estimated post-

calibration variances could not be evaluated properly; 3) no joint efforts were undertaken by the scientific

community as a whole to consistently disseminate and use/advance these test cases in future studies; 4) the

tests were strongly idealized and contained only a limited number of variants, which were not sufficient

to assess and compare methods under a broader set of conditions.

Most other comparison studies are smaller in scope. For example, Keidser and Rosbjerg (1991) and

Kuiper (1986) compared four and three inverse methods, respectively, and Sun et al. (2009) compared

four deterministic Kalman filters. A significant number of works rely on comparisons between only two

(with a maximum of three) inverse methods when proposing a new approach. Most remarkably, new

EnKF-variants are typically compared against the classical EnKF (e.g., Li et al., 2015; Gharamti et al.,

2015; Liu et al., 2016), but are seldom tested against other improved versions of the EnKF. Xu and Gómez-Hernández (2015) compared inverse sequential simulation and the sequential normal-score EnKF. Hendricks Franssen and Kinzelbach (2009) compared the EnKF and sequential self-calibration. Keating et al. (2010) compared null-space Monte Carlo and the MCMC-based DREAM-algorithm. Berg and Illman (2015) compared various methods for estimating hydraulic conductivities, including two inverse methods. Nowak (2009) compared the so-called Kalman ensemble generator to the quasi-linear geostatistical method by Kitanidis (1995). Keller et al. (2018, 2021) compared seven EnKF-based inverse algorithms, in both studies using as reference (different) synthetic realities. Keller et al. (2018, 2021) repeated data assimilation experiments with different ensemble sizes, including very large ensemble sizes, and also different repetitions for the same ensemble size (just varying the random seed used to generate the ensemble members). This increases the robustness of the comparison but is still based on a comparison with a synthetic reality. All of these listed comparison studies were important but with similar limitations as the two large comparison studies by Zimmerman et al. (1998) and Hendricks Franssen et al. (2009).

The major limitation is that almost all of these comparison studies were ad-hoc comparisons to data-generating synthetic realities - they did not compare with accurate reference solutions to the inverse problem. Without the latter, one can only compare two different approximations against each other and speculate about the correctness of both estimates. It is incorrect to ask inverse modeling results to be as close as possible to a numerically generated synthetic truth, unless in the limit of infinite information that is never reached in reality. Thus, the resulting estimate has to be a specific compromise between the smoothness



of prior assumptions and the limited amount of information contained in the available data. For the same reason, it is inadequate to require the post-calibration uncertainty to be as small as possible and celebrate a competitor that pretends a smaller estimation variance as the winner. Instead, there is an exact level

of post-calibration uncertainty that is justified by the weakness of prior information combined with the strength of information in the data. A remarkable exception is the study by Schöniger et al. (2012), who compared their normal-score version of the EnKF against a brute-force bootstrap filter. This is one of the very few studies where a high-accuracy, non-linearized reference solution was actually provided and used. However, the conditions in this study were tuned in an ad-hoc fashion (little data, relatively large

measurement error variance) so that it was possible to obtain that reference solution within a reasonable computing time.

The objective of our work is to overcome these shortcomings. We do so by providing a suite of well-defined benchmarking scenarios for stochastic inversion together with highly accurate reference solutions for the groundwater model inversion problem. As a starting point for this development, we choose a non-

linear groundwater model inversion of groundwater flow in multi-Gaussian log-conductivity fields. We take this choice because this is the most frequent type of scenario found in previous comparison papers. Later extensions to non-multi-Gaussian problems, to advective-diffusive transport, or to multiphase-flow are desirable. If our current study is successful, then researchers from the community will gladly use our benchmarking scenarios for assessing their new methods in future studies, and provide further extensions

to the benchmarking scenarios in a controlled and well-selected manner.



To achieve this goal, first, we discuss a suite of benchmarking metrics. This suite aims at capturing a fair trade-off between the various properties that a groundwater model inversion algorithm could have (e.g., accuracy under increasing non-linearity, speed, non-intrusiveness, ease of implementation, to name a few). Then, we propose a set of benchmarking scenarios, hoping that they will find widespread use as a

standard in the community in future studies. We also select and quickly summarize a recent MCMC-based algorithm that is capable of computing high-end reference solutions to these benchmarking scenarios, even if at high computational expense. With that algorithm, we produce and then present the reference solutions. Both the algorithm and the reference solutions are made available publicly for download. The reference algorithm is the pCN-PT MCMC algorithm recently proposed by Xu et al. (2020) for that very

purpose.

To provide a guiding example of how to use these benchmarking scenarios for comparisons, we perform and demonstrate a blueprint comparison between a specific EnKF variant and our reference solutions in our benchmarking scenarios. We do this with our discussed set of benchmarking metrics. When future studies add new algorithms to the literature and consistently use these scenarios, solutions, and metrics,

then an inter-comparable body of benchmarking studies will build up over time. With this, we hope to assist the groundwater model inversion community to move on faster and more efficiently in their research work. That means, we cordially welcome the scientific community to apply these benchmarking scenarios and reference solutions and evaluate their candidate inverse modeling methods in a multi-objective manner that will fairly and transparently reveal trade-offs between computational intensity, achievable

accuracy, (non-)intrusiveness to forward simulation codes, robustness against non-linearities and limits of

applicability posed by more or less restrictive assumptions.

Accordingly, this paper is organized as follows. In the methods section, we recapitulate the Bayesian

groundwater model inversion problem. Then, we discuss benchmarking metrics. Next, we define a base

domain for the benchmarking scenarios and provide five scenario versions. To conclude, we present our

choice of reference algorithm and present the reference solutions. In the application section, we quickly

summarize the EnKF version we are testing as a demonstrator example and then perform the benchmark-

ing comparison as a blueprint example of how to discuss the metrics and results in future studies. We

finalize our study with a discussion and conclusion on the unified benchmarking idea.

## 2 Problem definition: Bayesian groundwater model inversion

Given a forward model (here: the groundwater flow equation) $\boldsymbol{y} = M(\boldsymbol{\theta})$ with a vector of uncertain pa-

rameters (here: hydraulic log-conductivity) $\boldsymbol{\theta}$, and the vector of simulated measurements (here: hydraulic

head values) $\boldsymbol{y}$. The relation to the measured data $\boldsymbol{d}$ to be used for inversion is commonly written as

$$\boldsymbol{d} = M(\boldsymbol{\theta}) + \boldsymbol{e}, \tag{1}$$

where $\boldsymbol{e}$ represents the lump sum of unavoidable measurement errors and model errors. In the context of

numerical modeling, the parameter vector $\boldsymbol{\theta}$ contains discretized values of a random space function, such

as hydraulic log-conductivity values on some numerical grid.





The purpose of groundwater model inversion is to calibrate this model, i.e. to learn about $\boldsymbol{\theta}$ by matching $\boldsymbol{y}$ with $\boldsymbol{d}$. As typically the number of uncertain, discretized values in $\boldsymbol{\theta}$ is larger than the number of data values available in $\boldsymbol{d}$, the best one can do is to *estimate* $\boldsymbol{\theta}$. Conventionally, this is done by providing a

best-fit estimate under the regularization of geostatistical information about the spatial variability in $\boldsymbol{\theta}$. The next best thing one can do is to equip this estimate with a statement of uncertainty, e.g., via an estimation variance. The fully Bayesian approach provides a so-called posterior distribution of the uncertain parameters $\boldsymbol{\theta}$ given statistical prior knowledge about $\boldsymbol{\theta}$ and the measurement data $\boldsymbol{d}$. The posterior distribution is a full probabilistic description of the post-calibration uncertainty using a joint, multivariate

probability distribution of all model parameters.

The posterior distribution of the uncertain parameters $\boldsymbol{\theta}$ can be obtained through Bayes' rule as (e.g., Congdon, 2003)

$$p(\boldsymbol{\theta}|\boldsymbol{d}) = \frac{p(\boldsymbol{\theta})p(\boldsymbol{d}|\boldsymbol{\theta})}{p(\boldsymbol{d})} \propto p(\boldsymbol{\theta})p(\boldsymbol{d}|\boldsymbol{\theta}), \tag{2}$$

where $p(\boldsymbol{\theta})$ is the prior distribution of the unknown (hydraulic) parameters $\boldsymbol{\theta}$, which statistically describes

the prior knowledge of $\boldsymbol{\theta}$ before collecting any data $\boldsymbol{d}$; $p(\boldsymbol{d}|\boldsymbol{\theta})$ is the likelihood function, which quantifies the probability of the data $\boldsymbol{d}$ for a given realization of the uncertain parameters $\boldsymbol{\theta}$; and $p(\boldsymbol{d})$ is the marginal likelihood, which is a normalizing factor for the posterior distribution. This factor can often be ignored (hence the last part of equation (2)) when making inference of parameters (e.g., Duijndam, 1988; Cary and Chapman, 1988).



In our benchmarking scenarios, we will consider the residuals $e$ to be independently Gaussian with zero

mean. This leads to a likelihood function $p(d|\theta)$ that follows the Gaussian distribution over residuals $e$.

Given the zero-mean (unbiasedness) assumption, one merely needs to specify a variance for the errors,

and the definition of likelihood is complete. Separating between observational errors and model errors,

and speculating over their correlation or other form of dependence is beyond our scope of study (see, e.g.,

Schoups and Vrugt (2010)).

As statistical prior, we will consider $p(\theta)$ to be populated with log-hydraulic conductivities that are

multivariate Gaussian and second-order stationary, i.e. it suffices to specify a value for the mean and a

spatial covariance that depends on lag distances between locations on the numerical grid.

As a general property of the Bayesian inverse problem in this situation, the posterior distribution $p(\theta|d)$

would again be multivariate Gaussian if (and only if) the forward model $y = M(\theta)$ was linear in $\theta$, addi-

tionally to the Gaussian assumptions on errors and parameters. Only in that case, the posterior distribution

would be fully specified by a posterior mean and by a posterior (co-)variance. In typical applications, how-

ever, the forward model is non-linear in its parameters (like the groundwater flow equation), and therefore

we are interested in solving the full Bayesian problem. The best estimate and its variance is still a good

visualization of key aspects, but other aspects of the entire multivariate posterior distribution may be

important just as well.



# 3 Information and benchmarking metrics to be provided

To fairly evaluate the overall properties, performance, and fitness of a candidate inversion method, specific

fundamental information about the method and implementation must be provided, its solutions must be

compared to the reference solution, and user-relevant and application-relevant metrics and properties are

to be assessed and reported. Ideally, these are the same metrics and properties across all tested candidate

methods, even across different studies, so that inter-comparable results can accumulate to an overall body

of knowledge. Thus, we propose a range of statistics and metrics, see Table 1.

**Table 1.** Overview of information and metrics to be provided for the candidate inverse method

| Subsection | Category | Name | Equation |
|---|---|---|---|
| 3.1 | Methods & implementation | 1. Assumptions, simplifications (text) | (-) |
| | | 2. Ease of implementation (point-scale) | (-) |
| | | 3. Code FAIRness (point scale) | (-) |
| 3.2 | Benchmark selection | Reasons for choosing benchmarking scenarios (text) | (-) |
| 3.3 | Simulation verification | Assessing the plausibility of inversion results (text) | (-) |
| 3.4 | Metrics: accuracy | 1. Mean absolute error ($NMAE^*$) [1] | (5,3) |
| | | 2. Root mean square error ($NRMSE^*$) [2] | (7,3) |
| | | 3. Averaged KS Distance ($AD_{KS}$) | (10) |
| | | 4. Energy distance ($ND_E$) [3] | (13) |
| | *(optional)* | 5. Potential scale reduction factor ($\hat{R}$) | (17) |
| 3.5 | Metrics: compute cost | 1. Log-number of forward calls ($LN_f^*$) [4] | (-,3) |
| | | 2. Percent overhead wallclock time ($T_o$) | (18) |
| | | 3. Multi-core/multi-thread speedup loss ($S_\%$) | (19) |

[1] The mean absolute error $MAE$ needs a normalization ($NMAE$) to put it onto an interpretable scale, and an additional transformation ($NMAE^*$, Equation 3) to ensure it is bounded to $[0,1]$ like all other metrics.

[2] The root mean square error $RMSE$ needs a normalization ($NRMSE$) to put it onto an interpretable scale, and an additional transformation ($NRMSE^*$, Equation 3) to ensure it is bounded to $[0,1]$ like all other metrics.

[3] The energy distance $D_E$ needs a normalization ($ND_E$), but no additional transformation to ensure it is bounded to $[0,1]$ like all other metrics.

[4] The log-number of forward model calls $LN_f$ requires an additional transformation ($LN_f^*$, Equation 3) to ensure it is bounded to $[0,1]$ like all other metrics.

For obtaining easily interpretable visual diagnostics over a bundle of metrics, we ensure that all metrics

fall into the interval $[0, 1]$, where $0$ means best (no error) and $1$ means worst. This can be achieved, in most

cases, by meaningful scaling or normalization.

All five categories are discussed in the upcoming sections.

### 3.1 Methods and implementation

To assess the benchmarking performance later on and ensure the reproducibility of results, it is helpful to

first recall the fundamental assumptions, equations, and specific numerical settings chosen for the method:

1. List of assumptions and/or simplifications inherent in the method.

   Usually, each method has its own assumptions and/or simplifications when performing groundwater

   model inversion. For instance, EnKF methods assume the usefulness of correlations in inversion,

   which is related to an implicit global linearization (Nowak, 2009). Listing the assumptions and/or

   simplifications of the candidate method is helpful to get a more complete understanding of its per-

   formance.

2. Ease of implementation/intrusiveness into forward code. To our knowledge, this aspect can be gener-

   ally divided into 3 different levels, and so we suggest expressing it on a discrete point scale between

   zero and one:

   (a) Perfectly non-intrusive: Just forward calls with parameter modification (value: 0);





(b) Modified forward calls with different initial and boundary conditions as in adjoint-state sensitivities, but they can be controlled by simple adapted functions calls (value: 0.5);

(c) Actual need to enter the source code of the forward problem. Examples are dedicated adjoint computations, or stochastic moment equations that need new types of computations not foreseen in the original forward code (value: 1).

3. Results should always be easily reproducible by the scientific community. Therefore, we encourage authors to make not only their data FAIR (findable, accessible, interoperable, reusable, Wilkinson et al., 2016)), but also to apply the same principles to their software (Lamprecht et al., 2020; Martinez et al., 2019). If none of the following standards are met, a score of 1 is given; for each point met, the score reduces by 0.25:

(a) Make your software accessible online and explain how to set up an environment to run the code. Make versions that were used to generate published results explicit and assign a DOI to them.

(b) Meet domain-relevant community standards of your documentation.

(c) Output data follow FAIR principles (types and formats).

(d) Licenses (see, e.g., www.choosealicense.com, 2023, for an overview of open source licenses) are valid (dependent on the dependencies) and clearly stated.

## 3.2 Benchmark selection

What are the reasons to choose all, or only a subset of, the benchmarking scenarios? Although all the benchmarking scenarios are chosen carefully for a relatively fair and complete comparison, one may not use all the benchmarking scenarios due to some reasons. For example, if the candidate method is designed to solve problems with strong heterogeneity or large observational errors, then running benchmarking scenarios for mild heterogeneity is optional.

## 3.3 Simulation validation

We propose to show results and discuss some standard views on inversion results to allow domain experts to assess the solution. For this, we suggest plotting the spatial distribution of log-hydraulic conductivity via $\mathbb{E}(\ln(K))$ and the corresponding inversion uncertainty via $\mathrm{std}(\ln(K))$, and the same for pressure with $\mathbb{E}(p)$ and $\mathrm{std}(p)$. When using more than one benchmarking scenario, we recommend appending the respective plots after the first presented set of plots.

## 3.4 Metrics for accuracy

Accuracy means how close an approximate solution is to the true (reference) solution of the Bayesian inverse problem. The most comprehensive metric would address some distance (hopefully small) between the joint, high-dimensional posterior distributions of the discretized random space function of the candidate method and that of the reference solution. Since this is computationally unfeasible, we propose a





selection of different benchmark metrics, each of which highlights specific aspects that should be met by

235 the inversion method.

Recall that we wish to have all metrics in the interval $[0, 1]$. If a metric $\mathrm{METRIC}$ cannot be bounded

even after scaling (i.e. things worse than a score of $1$ cannot be excluded), it can be transformed for

visualization in a non-linear fashion that achieves a *visually* bounded representation:

$$\mathrm{METRIC}^*(x) = \frac{\mathrm{METRIC}}{1 + \mathrm{METRIC}}. \tag{3}$$

This transformation is almost linear for $\mathrm{METRIC} \ll 1.0$ and then saturates for larger values, thus limiting

the range of the transformed metric $\mathrm{METRIC}^*$ to $[0, 1]$. These transformed metrics will be marked by an

asterisk as superscript, e.g., $\mathrm{METRIC}^*$ instead of $\mathrm{METRIC}$ for a bounded normalized mean square error.

Overall, we suggest the following five metrics (plus their variations):

1. Mean absolute error ($\mathrm{MAE}$) across point-wise inversion statistics

$$\mathrm{MAE}(\hat{X}) = \frac{1}{N} \sum_{i=1}^{N} |\hat{X}_{cand,i} - \hat{X}_{ref,i}|, \tag{4}$$

where $N$ is the number of discrete points (nodes) in the spatial domain. $\hat{X}_{cand,i}$ is a posterior statistic

of $X$ at node $i$ provided by a candidate method for inversion, and $\hat{X}_{ref,i}$ is the corresponding posterior

statistic of $X$ at node $i$ in the reference solution. The featured variable $X$ could be hydraulic head





or hydraulic conductivity. The statistic $\hat{X}$ to be compared can be either the best estimate (posterior

mean) or the corresponding uncertainty (posterior standard deviation).

Being the (spatial) $\ell_1$ norm (rather than $\ell_2$), it is relatively mild on strong outliers at individual spatial

positions. For meaningful normalization to a range between 0 (perfect identity between solutions)

and 1 (largest meaningfully expectable error), we define:

$$\mathrm{NMAE}(\hat{X}) \;\; = \;\; \frac{\mathrm{MAE}(\hat{X})}{\mathrm{MAE}(\hat{X}_{prior})}, \tag{5}$$

$$\mathrm{MAE}(\hat{X}_{prior}) \;\; = \;\; \frac{1}{N}\sum_{i=1}^{N}|\hat{X}_{prior,i} - \hat{X}_{ref,i}|, \tag{6}$$

where $\hat{X}_{prior,i}$ is the same statistic as $\hat{X}_{cand,i}$ and $\hat{X}_{ref,i}$, but assessed based on the prior distribution

of the same variable $X$. Thus, $\mathrm{NMAE}(\hat{X}) = 0$ means a solution as accurate as the reference solu-

tion, and $\mathrm{NMAE}(\hat{X}) = 1$ means that the candidate inverse method is no better than to perform no

inversion at all. As it cannot be excluded that some inversion method could produce results worse

than the prior, we will use the non-linear transform from equation 3 to obtain $\mathrm{NMAE}^{*} \in [0,1]$.

2. Root mean square error ($\mathrm{RMSE}$) on point-wise inversion statistics

$$\mathrm{RMSE}(\hat{X}) = \sqrt{\frac{1}{N}\sum_{i=1}^{N}(\hat{X}_{cand,i} - \hat{X}_{ref,i})^2}, \tag{7}$$

with all symbols already defined above.





Being the (spatial) $\ell_2$ norm, the RMSE penalizes individual outliers against the reference solution

more strongly than the MAE. Again for normalization, we define:

$$\mathrm{NRMSE}(\hat{X}) \;=\; \frac{\mathrm{RMSE}(\hat{X})}{\mathrm{RMSE}(\hat{X}_{prior})}, \tag{8}$$

$$\mathrm{RMSE}(\hat{X}_{prior}) \;=\; \sqrt{\frac{1}{N}\sum_{i=1}^{N}(\hat{X}_{prior,i} - \hat{X}_{ref,i})^2}, \tag{9}$$

so that $\mathrm{NRMSE} = 0$ implies a fully accurate solution and $\mathrm{NRMSE} = 1$ again means that the solution

provided by the candidate method is no better than performing no inversion at all. Again, we will

use the bounded version, i.e. $\mathrm{NRMSE}^*$.

3. Kolmogorov–Smirnov (K-S) distance $D_{KS}$ on point-wise posterior distributions

$$D_{KS}(X_{cand,i}, X_{ref,i}) = \sup_{t}|F_{X,cand,i}(X) - F_{X,ref,i}(X)|, \tag{10}$$

where $F_{X,(\cdot)}$ denotes the (empirical or theoretical) cumulative distribution function (CDF) of a variable of interest $X$ at a specified node $i$ of interest.

The K-S distance measures the difference between two univariate probability distributions. It assesses the similarity between the posterior univariate distribution of an inferred variable (e.g., hydraulic conductivity, hydraulic heads) at an individual node $i$ by the candidate method $(F_{X,cand,i}(X))$ and the same by the reference solution $(F_{X,ref,i}(X))$. The K-S distance is the largest (over all possi-





ble values of a variable) difference between the two empirical or theoretical CDFs, and so is naturally

bounded between zero (perfect match) and one. The latter occurs when the compared distributions

allocate probabilities to entirely different value ranges of the variable under investigation.

The K-S distance can be plotted as a map (i.e. for each node $i$), and then spatially aggregated by

taking the spatial mean across the domain. Thus, we get as the averaged K-S distance $AD_{KS}(X)$ for

a quantity of interest $X$:

$$AD_{KS}(X) = \frac{1}{N} \sum_{i=1}^{N} D_{KS}(X_{cand,i}, X_{ref,i}), \tag{11}$$

which is again naturally bounded between zero (when the compared distributions are entirely identical at all averaged locations) and one.

4. Energy distance $D_E$ for global assessment of posterior distributions

While we use the K-S distance from above as a univariate tool (and make it global only via averaging), we can use the energy distance for a full, joint (multivariate) assessment of the posterior distribution. It serves well for that purpose because it does not require building a multivariate (empirical) distribution as the K-S distance would. The energy distance is defined as:

$$D_E(X_{cand}, X_{ref}) = (2\mathbb{E}\|X_{cand} - X_{ref}\| - \mathbb{E}\|X_{cand} - X'_{cand}\| - \mathbb{E}\|X_{ref} - X'_{ref}\|)^{1/2}, \tag{12}$$

where $\mathbb{E}$ denotes the expectation, $\|\cdot\|$ denotes the Euclidean norm (here: over the spatial domain, essentially a root mean square error), and the primed quantities are independent and identically





distributed copies of the non-primed quantities. Being a proper distance, it has a lower bound of zero

when the compared distributions are entirely identical. Although it has no natural upper limit, it can

be easily normalized as

$$ND_E(X_{cand}, X_{ref}) = \left( \frac{2\mathbb{E}\|X_{cand} - X_{ref}\| - \mathbb{E}\|X_{cand} - X'_{cand}\| - \mathbb{E}\|X_{ref} - X'_{ref}\|}{2\mathbb{E}\|X_{cand} - X_{ref}\|} \right)^{1/2}. \tag{13}$$

This normalized energy distance is always between 0 and 1, therefore there is no need to use any

transformation.

5. Potential scale reduction factor (PSRF) $\hat{R}$

The PSRF $\hat{R}$ (Gelman et al., 1992) is defined for a set of $m$ Markov chains, each of which has $n$

samples. The within-chain variance (for each node separately) is estimated as

$$W = \frac{1}{m(n-1)} \sum_{j=1}^{m} \sum_{i=1}^{n} (X_j^{(i)} - \bar{X}_j)^2, \tag{14}$$

where $X_j^{(i)}$ is the $i$th sample of the $j$th chain and $\bar{X}_j$ is the mean of the samples in the $j$th chain (for

each node separately). The between-chain variance (for each node separately) is estimated as

$$B = \frac{n}{m-1} \sum_{j=1}^{m} \left( \bar{X}_j - \frac{1}{m} \sum_{j=1}^{m} \bar{X}_j \right)^2. \tag{15}$$





Then, the estimated variance $V$ is a weighted average of the within-chain variance and between-chain

variance:

$$V = (1 - \frac{1}{n})W + \frac{1}{n}B. \tag{16}$$

Finally, the PSRF $\hat{R}$ (for each node separately) is defined as

$$\hat{R} = \sqrt{\frac{V}{W}}, \tag{17}$$

and we obtain a global metric by averaging it across all spatial nodes, i.e., the average $\mathrm{PSRF}$ $\bar{R}$.

This metric serves to test a method against itself under repeated executions. It applies to methods that

contain random components, such as MCMC methods. We will mainly use it to establish and discuss

the accuracy of our reference solutions. As recommended by Gelman et al. (2013) and Brooks and

Gelman (1998), the potential scale reduction factor $\hat{R}$ should be lower than 1.2 so that one can claim

to achieve convergence and fully explore the target posterior. One could, in principle, also use the

$\hat{R}$ to compare $m = 2$ methods rather than $m$ chains. We will not do so because equations 14 to 17

reveal that $\hat{R}$ is a variant of $\mathrm{MAE}$, just scaled differently for a different purpose.

We refrain from including correlation maps in our set of metrics. With correlation maps, we refer to

cross-covariance fields between a quantity of interest at one spatial location and some other quantity of in-

terest at all possible spatial locations. An example would be the cross-covariance between hydraulic heads



at a pumping well location and the hydraulic log-conductivity throughout the domain. We do not include

such maps, because different methods provide either prior or posterior covariances: MCMC-type methods

would provide posterior covariances; QLGA-type methods would provide prior covariances (linearized

about the $\mathrm{MAP}$); EnKF-type methods would provide prior covariances before the first data assimilation

step and conditional covariances during/after assimilation, and iterative/sequential EnKF-type methods

would provide intermediate iterates.

### 3.5  Metrics for computational effort

A high accuracy, as measured with the metrics introduced in Section 3.4, might come at high computing

costs. Very accurate methods may be too computationally demanding for larger-scale problems. There-

fore, the trade-off between accuracy and computing costs provided by different methods is important. We

use a total of three metrics of computing cost:

1. The number of forward calls $N_f$ in the inversion, is relatively easy to evaluate. We propose to look

   at this number as an order of magnitude. For example, it could be around 100 model evaluations for

   sequential self-calibration using the adjoint state method and Gauss-Newton optimization, but 1000

   if gradient descent is used instead of Gauss-Newton. It could be 100-1000 model evaluations for

   EnKFs and $\approx 10^5$-$10^6$ model evaluations for efficient MCMCs. To consider the logarithmic scale,

   we use $\mathrm{LN}_f = \log_{10}(N_f)$, and then apply equation 3 to obtain $\mathrm{LN}_f^*$ between zero (no computing

   effort) and one (infinite computing effort).





2. The second metric is percent overhead wallclock time $T_o$. Wall-clock time is the actual time elapsed while the code runs. We define overhead wallclock time $T_o$ as wallclock time $T_w$ minus the time spent on calls to the forward model $T_f$. Finally, we take the fraction:

$$T_o = \frac{T_w - T_f}{T_w},$$ (18)

and realize that $T_o$ is automatically in $[0,1]$. As wallclock time aspects can depend on the used machine (although many machine influences would cancel out in the above normalization), technical machine details have to be reported.

3. Multi-core/multi-thread speedup loss $S_\%$: If one took $N$ times the number of compute nodes, one could ideally be $N$ times faster. With mutually waiting algorithms, real-life implementations, and real hardware, the achievable speedup factor is $K \leq N$. As speedup loss, we define:

$$L_{s\%} = 1 - \frac{K}{N},$$ (19)

and $L_{s\%}$ is automatically between zero (for an ideal, lossless speedup at perfect scaling) and one (for absolutely no speedup although using $n \to \infty$ compute nodes). Report this metric if (and only if) you tested parallel scaling on dedicated parallel hardware. If so, comment on the technical and programming aspects of parallelization. Also, as $L_{s\%}$ usually depends on the number $N$ of nodes used (scaling is typically not linear), $N$ must be reported. If not, it is optional to provide information about possible (future) parallelization options.





## 3.6 Suggested reporting and visualization

To ensure consistency in reporting and visualization, we strongly recommend that benchmarking results are reported in the same order as specified in Table 1. For user convenience, we also provide a Jupyter notebook, available for download [1]. It accesses the reference solutions as will be discussed in Section 5.3, and asks the user to provide access to the own candidate solution (to be benchmarked). Then, it automatically computes and plots the set of metrics for accuracy. It also offers the option to enter all other aspects discussed above, starting from *list of assumptions* in Section 3.1 up to *multi-core/multi-thread speedup loss* in Section 3.5.

We will provide our example exercise of benchmarking in Section 6 exactly in the recommended order, and using our provided notebook and its output figures.

## 4 Benchmarking scenarios and obtaining their reference solution

### 4.1 Governing equations, basic domain and forward model

In our benchmarking scenarios, the forward model $M(\boldsymbol{\theta})$ is given by steady-state or transient groundwater flow in 2D confined aquifers, depending on the exact scenario definition. The governing equation for fully saturated transient groundwater flow in confined aquifers is given in (Bear, 1972):

$$\nabla \cdot (K \nabla H) + W - S_s \frac{\partial H}{\partial t} = 0, \tag{20}$$

---

[1]https://github.com/LS3-university-of-stuttgart/hydrological-inversion-benchmarking





**Table 2.** Pumping wells used in benchmarking scenarios

| Well number | Grid position | Position $(m, m)$ | Pumping rate $(m^3/d)$ |
|---|---|---|---|
| #1 | (10,47) | (500, 2350) | 120 |
| #2 | (70,47) | (3500, 2350) | 70 |
| #3 | (40,71) | (2000, 3550) | 90 |
| #4 | (40,21) | (2000, 1050) | 90 |

where $S_s$ is specific storage [L$^{-1}$]; $H$ is the hydraulic head [L]; $t$ is time [T]; $\nabla \cdot$ is the divergence operator; $\nabla$ is the gradient operator; $K$ is the hydraulic conductivity [LT$^{-1}$]; and $W$ is the volumetric injection flow rate per unit volume of aquifer [T$^{-1}$]. In our scenarios, $K$ is scalar, i.e. we look at locally isotropic aquifers.

For steady-state flow, the third item on the left of the equation is equal to zero, and thus the steady-state

ground flow equation can be written as:

$$\nabla \cdot (K \nabla H) + W = 0. \tag{21}$$

Although Eq.20 and Eq.21 are originally for 3D flow, in this work they are applied to 2D flow conditions. Both flow regimes need boundary conditions. In all of our scenarios, we use a 5000 m × 5000 m domain. Its north and south boundaries are impermeable; its west and east boundaries have specified

heads equal to 20 m and 0 m, respectively. Scenarios with transient groundwater flow additionally need initial conditions, and we set the initial head to be uniform and equal to 20 m throughout the domain except for the east boundary. Regardless of the flow regime, we define four pumping wells for groundwater abstraction (Figure 1). The exact locations and pumping rates are provided in Table 2.





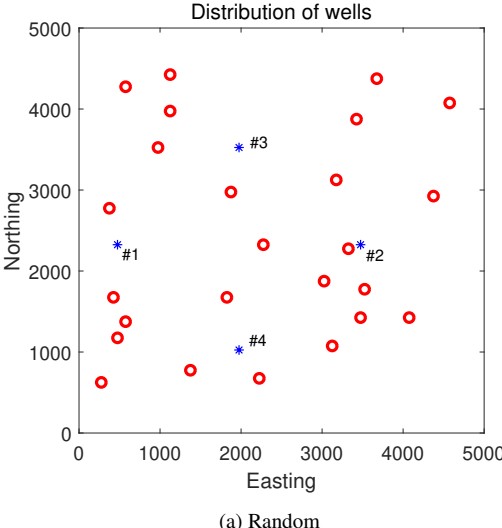

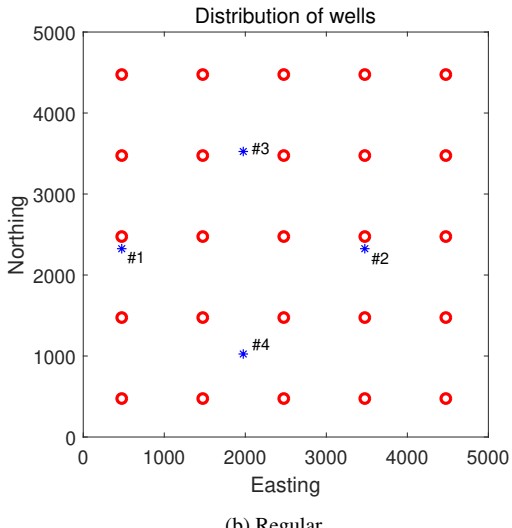

(a) Random

(b) Regular

**Figure 1.** Distribution of pumping wells (blue asterisks) and observation wells (red circles) in the benchmarking scenarios. The left figure shows a random distribution of observation wells (scenarios S0, S2, S3, S4); the right figure shows regularly distributed observation wells (scenario S1).

Both equations are solved by using the groundwater flow simulator MODFLOW (McDonald and Har-
390 baugh, 1988). We define a synthetic aquifer as a flow domain on a grid of $100 \times 100 \times 1$ cells, where each cell is 50 [L] by 50 [L] by 50 [L]. For transient simulations, the total simulation time is 10000 [T], evenly discretized into 10 time steps. All model setup files are published online (Xu, 2023).

### 4.2 Benchmarking scenarios

Overall, we define five different benchmarking scenarios based on the basic domain described above.
Good candidate inverse methods should be able to solve problems with different characteristics. There-
fore, our scenarios differ according to (1) groundwater flow (steady-state versus transient), (2) the distribu-
tion of observations (random versus regular), (3) the degree of heterogeneity that affects the nonlinearity



of the inverse problem, and (4) the magnitude of measurement error, which affects the convergence requirements of candidate inverse methods.

All scenarios share a mean value for log-hydraulic conductivity $\theta = \ln K$ of $\mu_\theta = -2.5$ [ln(LT$^{-1}$)]. Although they have different standard deviations for log-conductivity, all scenarios share the exponential covariance model with a slightly rotated anisotropy. It has correlation length parameters $\lambda_{max} = 2000L$ and $\lambda_{min} = 1500L$, and the dominant principal axis (the one for $\lambda_{max} = 2000L$) aligned from north-west to south-east. Also, across all scenarios specific storage is homogeneous and equal to 0.003 [L$^{-1}$].

The details of all scenarios are provided in Table 3. We deliberately do not include all possible combinations as scenario variations. Instead, we define a base scenario S0, and from there do "one-at-a-time" variations:

- S0 is the base case. It features a relatively strong degree of heterogeneity with $\sigma_\theta = 2$, relatively accurate measurement data with $\sigma_e = 0.05$ [L], irregularly placed observations, and steady-state groundwater flow.

- S1 features the regular grid of observations instead of the random one. While irregular monitoring networks are more realistic, the very close spacing of a few monitoring wells may pose a problem to some methods due to their high autocorrelation. Therefore, S1 is a fallback scenario.





**Table 3.** Definition of scenarios

| Scenarios | Std.dev of $\ln K$ synthetic truths | Std.dev of measurement errors | Well distribution | Flow state |
|---|---|---|---|---|
| S0 | 2 | 0.05 | Random | Steady |
| S1 | 2 | 0.05 | Regular | Steady |
| S2 | 1 | 0.05 | Random | Steady |
| S3 | 2 | 0.1 | Random | Steady |
| S4 | 2 | 0.05 | Random | Transient |

– S2 is again like S0, but reduces the strength of heterogeneity from $\sigma_\theta = 2$ to $\sigma_\theta = 1$. While $\sigma_\theta = 2$ is a more realistic degree of heterogeneity, it may already be challenging for methods that are explicitly or implicitly linearization-based. Therefore, S2 is a fallback scenario.

– S3 is again like S0, but increases the assumed level of observational errors from $\sigma_e = 0.05$ [L] to $\sigma_e = 0.1$ [L]. Given the overall head difference of 20 [L] across the domain by the boundary conditions, these values can be classified as high and medium accuracy, respectively. Iterative or sampling-based methods may have problems with the accuracy requirement posed by the large accuracy in S0. Once again, S3 is a fallback solution. However, as posterior uncertainties will remain larger for smaller measurement accuracy, S3 may also trigger stronger non-linearities across the larger remaining post-calibration uncertainty ranges.

– S4 changes S0 to feature transient (instead of steady-state) groundwater flow. This is relevant for EnKF-type methods that work via transient data assimilation and that do not iterate.





### 4.3 Synthetic cases for our benchmarking scenarios

We use the `sgsim` code, a sequential Gaussian simulation module of the `GSLIB` software (Deutsch and Journel, 1998), to generate two log-conductivity fields according to our benchmarking scenarios. We use them as data-generating synthetic truth fields, providing synthetic measurement data for inversion.

Therefore, we generate one realization with $\sigma_\theta = 1$ (scenario S2), and then simply scale it to obtain a realization for $\sigma_\theta = 2$ (scenarios S0, S1, S3, S4). The resulting synthetic fields are shown in Figure 2, and are called synthetic truth 1 (left) and synthetic truth 2 (right). All details of the parameters and covariance function settings are shown in Table 4. Please be aware that the goal of benchmarking is *not* to be as close to the synthetic fields as possible (with a posterior uncertainty as small as possible), but to be as close to

a high-end reference solution as possible (i.e. close in best estimate, posterior standard deviation and so forth).

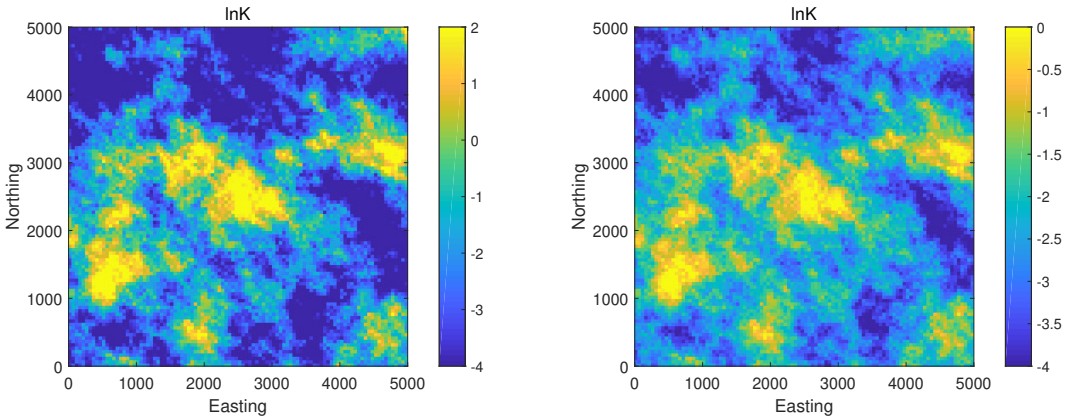

**Figure 2.** Synthetic truth of $\ln K$. Left: synthetic truth 1 with a standard deviation of 2 $\ln(L/T)$; Right: synthetic truth 2 with a standard deviation of 1 $\ln(L/T)$.





**Table 4.** Parameters of the random functions describing the heterogeneity of the two synthetic truth fields. $\lambda_{max}$ and $\lambda_{min}$ are the correlation ranges along the principal axes of orientation.

|  | Mean | Variogram type | $\lambda_{max}$ | $\lambda_{min}$ | Std. dev. | Angle |
|---|---|---|---|---|---|---|
| synthetic truth 1 | -2.5 | Exponential | 2000 | 1500 | 2 | 135 |
| synthetic truth 2 | -2.5 | Exponential | 2000 | 1500 | 1 | 135 |

There are 25 observation wells, where we simulate measurements for providing synthetic data values of hydraulic heads. Their spatial distribution is either random (scenarios S0, S2-S4) or regular (scenario S1). For all scenarios, the measurement errors follow Gaussian distributions with mean zero. The standard

deviation (std. dev.) varies between scenarios as shown in Table 3. After simulation with MODFLOW, we add randomly generated error values according to these distributions to obtain the synthetic data to be used for inversion. For the steady-state scenarios (S0 to S3), this results in 25 data values, and for the transient scenario (S4), this results in 10 time steps times 25 data values, i.e. in a total of 250 data values.

**4.4    Algorithm used for the reference solutions: the pCN-PT**

Markov chain Monte Carlo (MCMC) simulation is a widely-used sampling method to approximate the Bayesian statistical inverse solution (Tierney, 1994; Grandis et al., 1999; Schott et al., 1999; Smith and Marshall, 2008). In the literature, many MCMC algorithms have been proposed. The Metropolis–Hastings algorithm (Hastings, 1970) is a widely-used MCMC algorithm for standard cases. However, it has relatively low efficiency in reaching convergence, especially when encountering high-dimensional problems

(Haario et al., 2006). In the context of groundwater model inversion, the dimensionality relevant to MCMC

efficiency is equal to the number of discretization cells used for the random field. This is *not* to be confused with the spatial dimension of groundwater flow.

Later, several new methods were proposed to improve the efficiency and effectiveness of MCMC algorithms (Tierney and Mira, 1999; Haario et al., 2001, 2006; ter Braak, 2006). The preconditioned Crank Nicholson (pCN) algorithm (Cotter et al., 2013a) is an efficient MCMC algorithm with dimension robustness. In plain words, the pCN is tailored to automatically fulfill multi-Gaussian priors, so that spatial refinement does not affect its performance. Therefore, the acceptance probability for proposed solutions only depends on the likelihood of matching with the data.

However, for many hydrogeological inverse problems, an additional problem is that the posterior distribution is multimodal. To effectively sample such multimodal posterior distributions, parallel tempering (PT) (Altekar et al., 2004; Earl and Deem, 2005) is a good candidate. It runs multiple chains with different temperatures in parallel. The hot chains can more easily explore the whole parameter space since the likelihood of the hotter chains is flatter and broader as the temperature increases. The cold chains perform precise sampling in high-likelihood regions of the parameter space. With regular swaps between the members of hot and cold chains, the hotter chains can help the coldest chain with unit temperature (target chain) access the desired regions of the parameter space.

In this work, we use the pCN-PT algorithm by Xu et al. (2020) to get the reference solution. It combines the pCN algorithm with PT. We chose it because it is well-suited for high-dimensional and nonlinear problems, even when the posterior distribution is multimodal. Xu et al. (2020) has proven its capacity to



deal with both high-dimensional linear problems and high-dimensional nonlinear problems. The perfor-

mance in linear problems was tested in a synthetic study, and Xu et al. (2020) could be compared to exact

(analytical) reference solutions obtained by kriging. For the nonlinear cases, Xu et al. (2020) compared

multiple MCMC runs to assess convergence. Again, to work with reference solutions, they compared

it to a solution obtained by plain rejection sampling in a case with very weakly informative data. Fi-

nally, they showed that their pCN-PT is superior to the original pCN-MCMC for Bayesian inversion of

multi-Gaussian parameter fields. The algorithm for pCN-PT is provided in the appendix, and full details

can be found in Xu et al. (2020). Possible extensions for multi-facies aquifers with internal (Gaussian)

heterogeneity or to cases with uncertain covariance parameters exist (e.g., Xiao et al., 2021).

### 4.5 Generating the high-end reference solutions

Operating the pCN-PT requires a set of decisions to be taken. This mainly concerns tuning its parame-

ters such as the temperature ladder, the pCN jump size parameter per temperature, and the frequency of

between-chain swapping proposals. Many studies (e.g., Gelman et al., 1996; Roberts et al., 1997, 2001;

Predescu et al., 2005; Laloy et al., 2016) indicated that acceptance rates for MCMC-based methods be-

tween 10% to 40% perform close to optimal and that the optimal acceptance swap rates for parallel

tempering should range from 8% to 39%. Here, like in the study by Xu et al. (2020), we control the ac-

ceptance rate in a range from 20% to 30% and the swap acceptance rate in a range from 10% to 30% by

adjusting the jumping factor and temperature ladders.



Finally, we have to decide on the length of runtime. For scenarios S0-S4, we generate 800,000 MCMC

realizations by the pCN-PT, saving every 20th realization. The first half of all realizations is discarded as

a burn-in period. That means the reference solution for each benchmarking scenario consists of 400,000

MCMC steps, thinned out to 20,000 realizations. This thinning-out mostly serves to reduce memory re-

quirements. It has virtually no effect on the quality of the reference solutions, as it mainly reduces the

along-chain autocorrelation of the realizations. For later assessment of convergence, we repeat each sce-

nario four times in independent pCN-PT runs. These solutions are uploaded and made available publicly

on an institutional data repository (Xu, 2023).

## 5    Results: high-end reference solutions for the benchmarking scenarios

### 5.1    Solutions

To visualize the reference solutions to the benchmarking scenarios, Figures 3 and 4 show the maps of

mean and standard deviation of the $\ln K$ realizations obtained by the pCN-PT for all scenarios S0 to S4.

It can be seen that the reference solutions in all scenarios capture the main features of the synthetic fields.

As an additional plausibility check, comparing the results of S4 with those of scenarios S0-S3, one can

see that the more observations have been assimilated, the better identification with smaller uncertainty can

be achieved. Also, scenario S2 (with the smaller prior standard deviation compared to S0) has a smaller





posterior standard deviation, and scenario S3 (with the larger measurement error variance compared to

S0) has a slightly smoother mean.

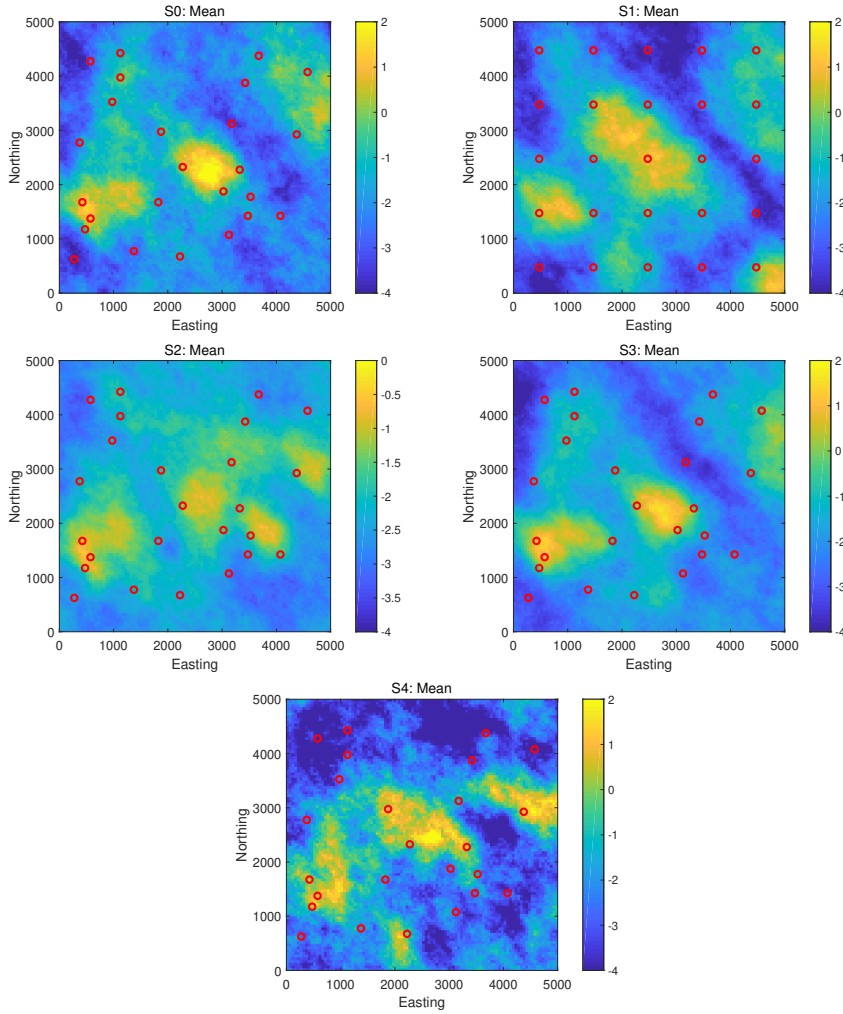

**Figure 3.** Scenarios S0-S4. The mean of $\ln K$ realizations obtained by the pCN-PT. The color scale used for scenario S2 is the same as that used for the synthetic truth 2 in Figure 2.

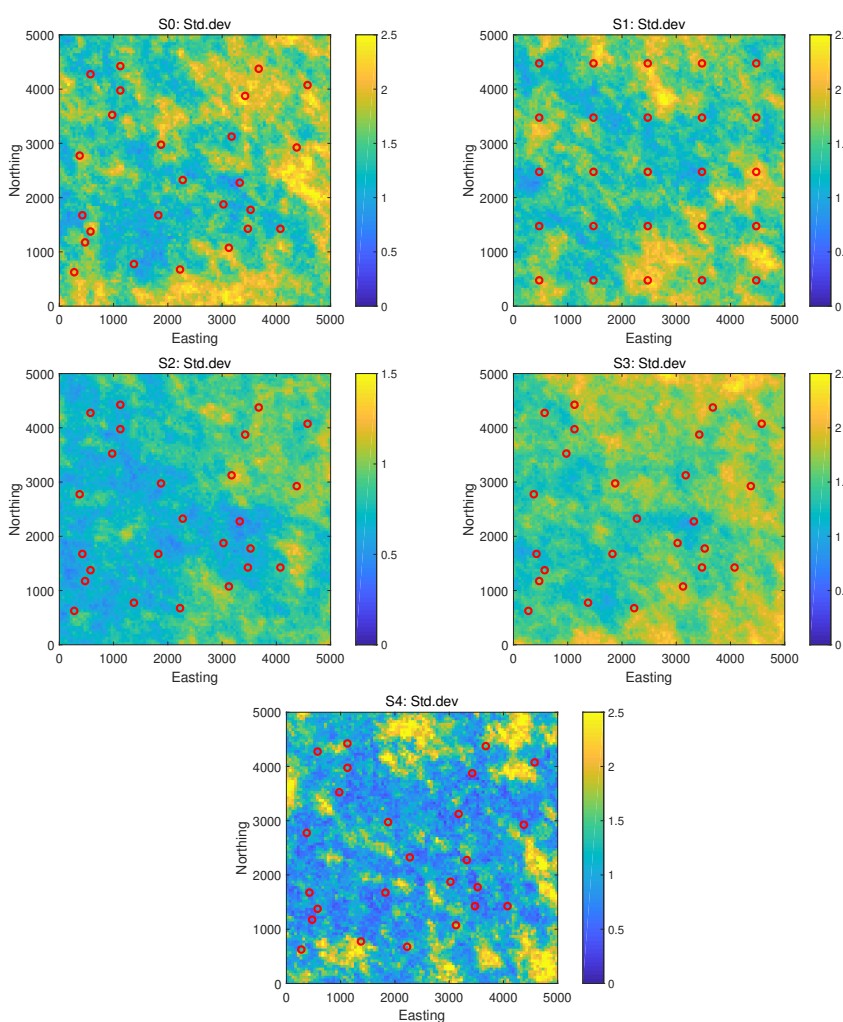

**Figure 4.** Scenarios S0-S4. The standard deviation of $\ln K$ realizations obtained by the pCN-PT. Scenario S2 is shown on a different color scale because it has only half the prior field variance.

## 5.2 Convergence assessment

For more quantitative assessment, Figure 5 shows the root mean square error (RMSE) of $\ln K$ against

the synthetic truth for scenarios S0-S4 along the runtime of the pCN-MCMC algorithm. The RMSE is

calculated according to Eq.(7) while swapping the roles of "candidate" against "reference" for "reference"

against "synthetic". To obtain the shown trace plots, these RMSE values are computed and recorded for

each step in the target MCMC chain. Recall that the expectation for a reference solution is *not* to yield

values as small as possible here, but to strike the correct compromise between prior information and

noise-affected data. We merely use the RMSE here to discuss aspects of the MCMC operation for our

reference solutions. It is observed that RMSE in all scenarios decreases sharply within a very short (and

hence hardly visible) initial period in the plots, and then converges quickly to a stable distribution along

the chains. The RMSE for scenario S2 (the one with the smaller prior standard deviation) is smaller than

for all other scenarios. This is because the more narrow prior leads to a more narrow posterior closer to

the data-generating synthetic truth, which is to be expected.

Figure 6 shows log-likelihood values in matching the synthetic data along the target MCMC chain for

scenarios S0-S4. The pCN-PT converges more quickly in the steady-state scenarios (S0 to S3) than in the

transient scenario (S4). The visible aspects of burn-in periods are very short in scenarios S0-S3. Due to

the more informative data in S4, the posterior is much more narrow and more shifted against the prior. The

more narrow posterior also requires smaller jump size factors in MCMC algorithms. Together with the



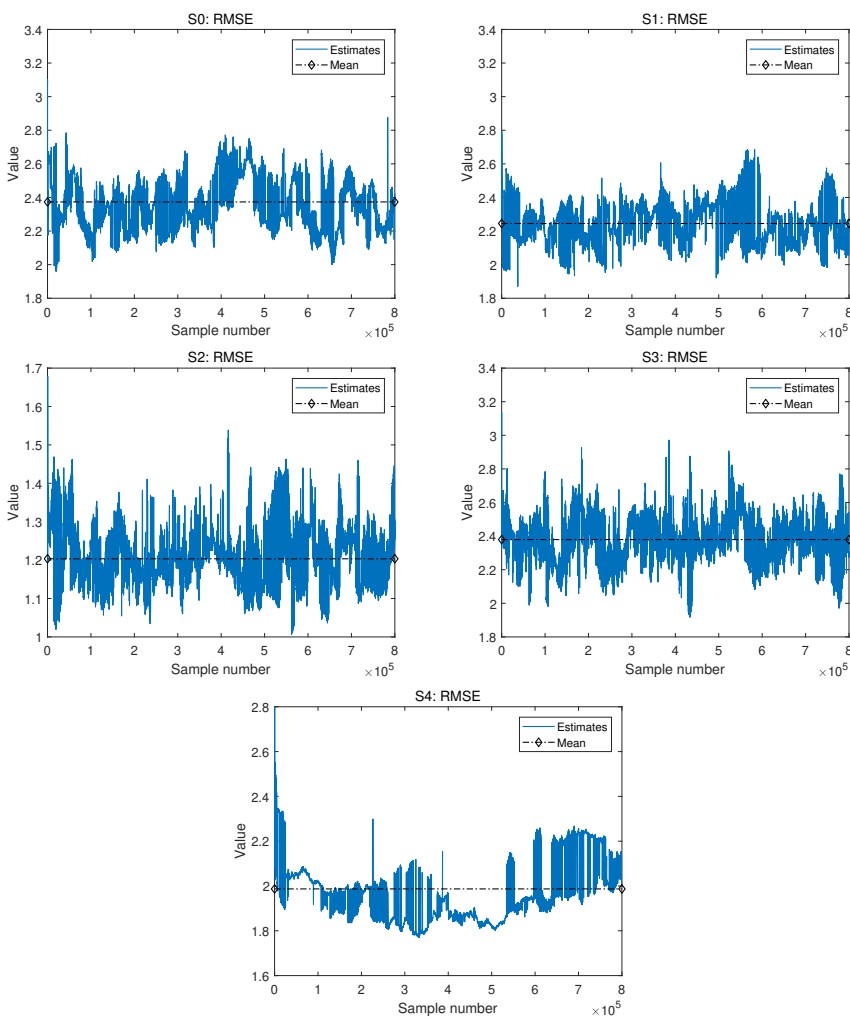

**Figure 5.** The RMSE error between the log K values of Scenarios S0-S4 of the pCN-PT solution vs. the synthetic MODFLOW-based truth over all samples. The blue line and black dash-diamond line in the figure correspond to the RMSE of the updated samples, and the mean of the RMSE, respectively.





more offset posterior, this results in a longer visible burn-in period. After the burn-in period, all scenarios

show a good exploration of realizations with a stable distribution of likelihood values.

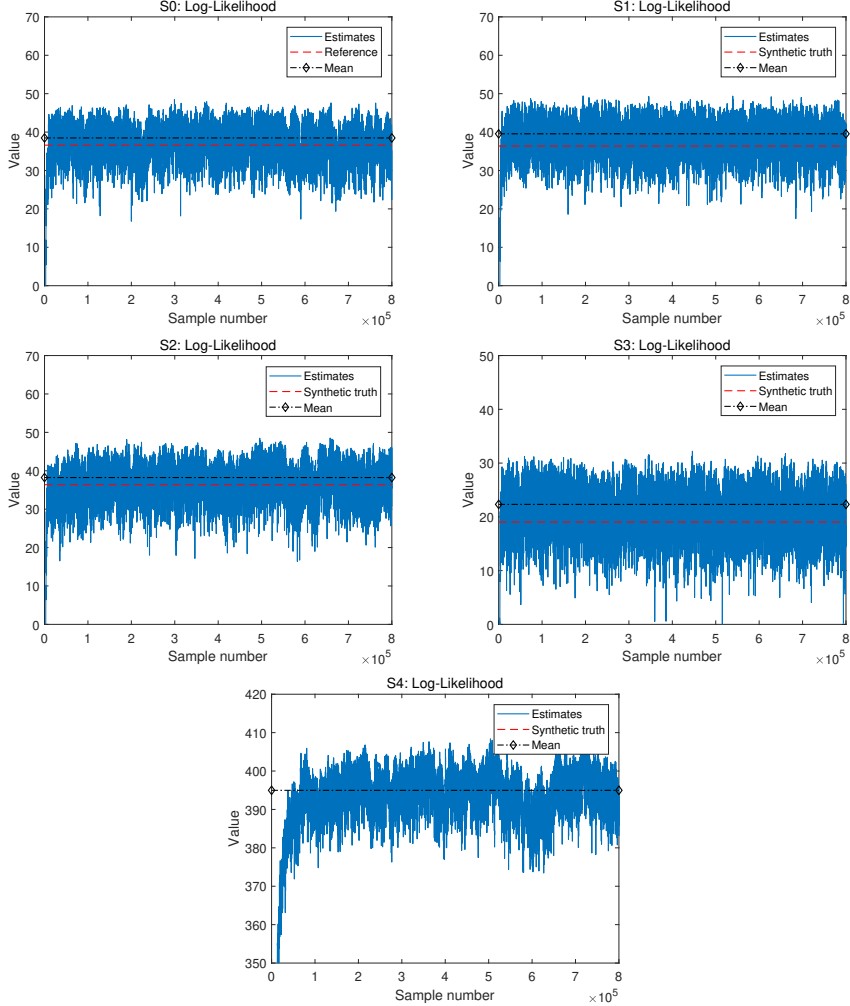

**Figure 6.** Scenarios S0-S4. Log-likelihood as obtained by the pCN-PT for the different cases. The blue solid line, red dashed line, and black dash-diamond line in the figure of the log-likelihood correspond to the log-likelihood of the updated samples, the reference, and the mean of the log-likelihood for the pCN-PT, respectively.

Figures 7 and 8 show the potential scale reduction factor $\hat{R}$ of the reference solutions for all scenarios.

Specifically, Figure 7 shows the evolution of the spatial maximum and the mean of $\hat{R}$ along the MCMC



runtime, and 8 shows the spatial pattern of the final $\hat{R}$ values at the end of MCMC runtime. Recall that the

recommended value for $\hat{R}$ is 1.2 or below. The figures show that we achieve values for the spatial average

$\bar{R}$ below 1.2 in all scenarios. For the spatial maximum of $\hat{R}$, i.e. the worst single pixel on the map, we

achieve values close to 1.2 in all scenarios but in scenario S4. This scenario is the transient one. Overall,

these results confirm the conclusions from before: the reference solutions in all scenarios have an excellent

quality (with scenario S4 falling behind and showing only a plain good quality); scenario S3 with its more

forgiving (larger) measurement error variance is easier to handle, converges faster, and so the values of

$\hat{R}$ drop below the threshold value of 1.2 more quickly and in larger parts of the spatial domain; scenario

S2 with the more narrow prior is only slightly easier to handle and converges only slightly faster than the

other ones; scenario S4 with the more informative data is harder to handle and it converges more slowly.

### 5.3    Accessing the high-end reference solutions

The reference solutions are published in an online data repository (Xu, 2023). There, prefixed with "ref_",

the reference chains of the different scenarios are accessible.

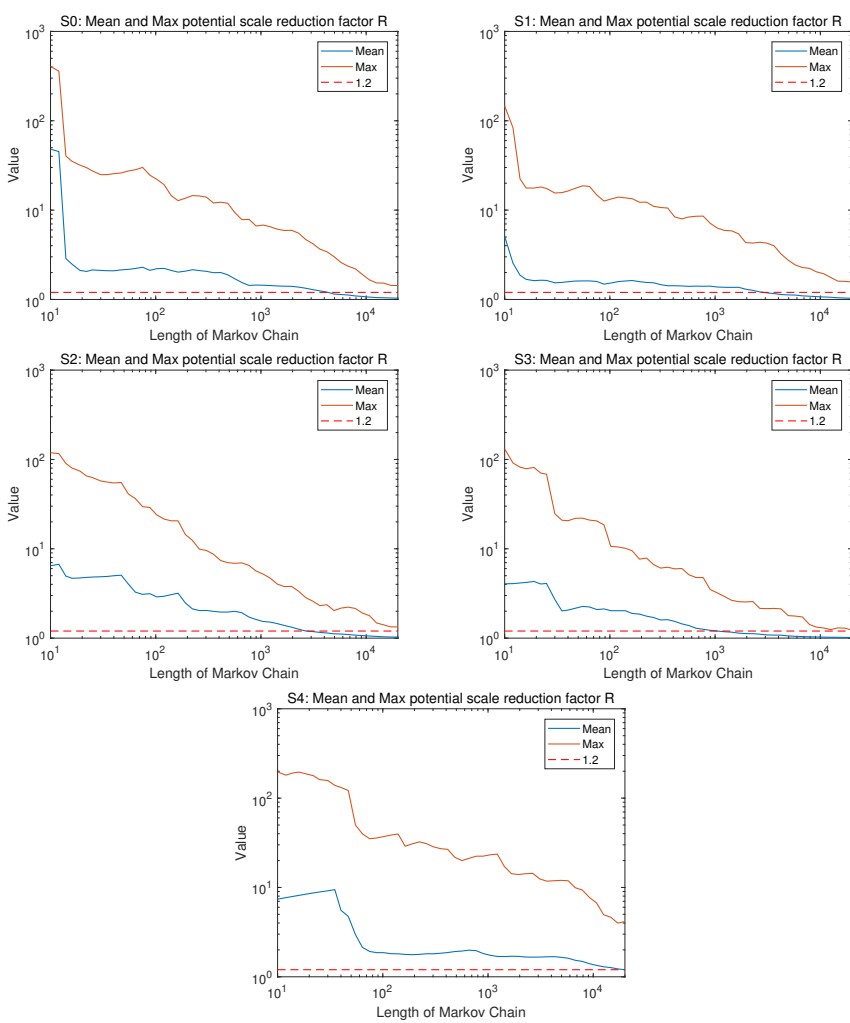

**Figure 7.** Scenarios S0-S4. The evolution of mean (blue) and maximum (red) potential scale reduction factor obtained by the pCN-PT. The red dashed line, the blue line, and the brown line in the evolution of mean and maximum potential scale reduction factor correspond to the value 1.2, the mean value, and the maximum value for posterior samples, respectively.

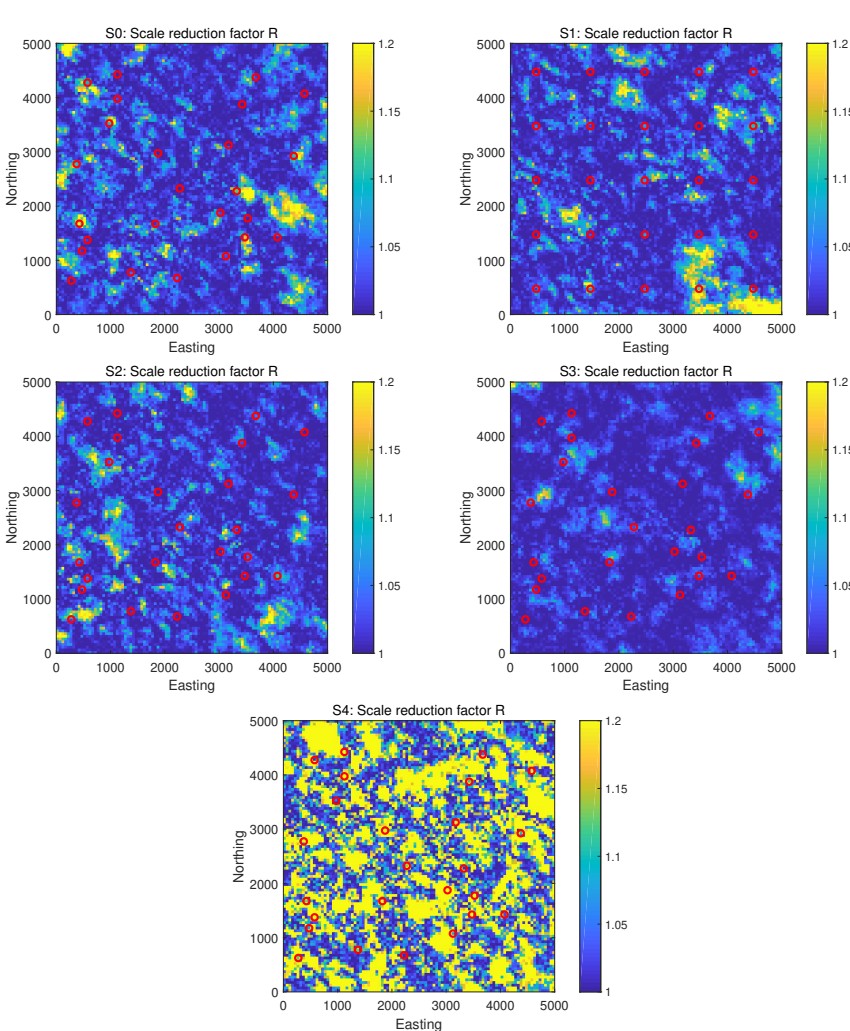

**Figure 8.** Scenarios S0-S4. Potential scale reduction factor field obtained by the pCN-PT.

## 6 Example application: testing the EnKF

### 6.1 EnKF: Methods and implementation

The EnKF was proposed by Evensen (1994), based on the Kalman filter (Kalman et al., 1960) and the extended Kalman filter (McElhoe, 1966), to better deal with nonlinear state-transfer functions. The Kalman

filter is an efficient recursive filter but can only handle linear problems. The extended Kalman filter can handle nonlinear state-transfer functions by linearizing nonlinear systems through a Taylor expansion. Still, it fails to handle large, strongly nonlinear systems due to the large storage, time consumption, and accumulative error induced by the linearization process (Xu et al., 2013a). To tackle data assimilation in large, nonlinear systems, the EnKF was proposed, in which an ensemble of realizations is used to ap-

proximate the (cross-)covariances during the analysis step, instead of propagating the (cross-)covariances using a linear(ized) state-transfer function as in the Kalman filter. As an efficient and effective inverse modeling approach for parameter estimation, the EnKF has received lots of attention and applications in many fields (e.g., Evensen, 2003; Bertino et al., 2003; Chen and Zhang, 2006; Aanonsen et al., 2009; Hendricks Franssen and Kinzelbach, 2008, 2009; Xu et al., 2013b; Xu and Gómez-Hernández, 2018).

The EnKF used here mainly contains the following steps:

1. Initialization step. An initial ensemble of hydraulic log-conductivity fields $\ln K_0$ is generated from the prior, here using sequential Gaussian simulation.





2. Forecast step. For the $i^{th}$ realization at the $t^{th}$ time step, the piezometric heads $\mathbf{H}_{i,t}$ are forecasted on the basis of the piezometric heads $\mathbf{H}_{i,t-1}$ and the log-conductivities $\mathbf{lnK}_{i,t-1}$ from the $(t-1)^{th}$ time step through a state-transition equation $\psi(\cdot)$ (see Eq.(22)), which is the transient groundwater flow equation:

$$\mathbf{H}_{i,t} = \psi(\mathbf{H}_{i,t-1}, \mathbf{lnK}_{i,t-1}) \tag{22}$$

3. Assimilation step. The log-conductivities $\mathbf{lnK}_{i,t}^{a}$ (a function of the log-conductivities at the last time step $\mathbf{lnK}_{i,t-1}^{f}$, the Kalman gain $\mathbf{G}_{i,t}$ and the misfit between the forecasted heads $\mathbf{H}_{i,t}^{f}$ and observed heads $\mathbf{H}_{t}^{o}$ at observation locations) are sequentially updated by assimilating observed piezometric heads $\mathbf{H}_{t}^{o}$ (Eq.(23)). The Kalman gain $\mathbf{G}_{i,t}$ is a function of the cross-covariances $\mathbf{C}_{\mathbf{lnK}_{i,t-1}\mathbf{H}_{i,t}}$ between log-conductivities and forecasted heads at observation locations, and auto-covariances $\mathbf{C}_{\mathbf{H}_{i,t}\mathbf{H}_{i,t}}$ of forecasted heads at observation locations (Eq.(24)).

$$\mathbf{lnK}_{i,t}^{a} = \mathbf{lnK}_{i,t-1}^{f} + \mathbf{G}_{i,t}(\mathbf{H}_{t}^{o} + e_{i,t} - \mathbf{H}_{i,t}^{f}) \tag{23}$$

$$\mathbf{G}_{i,t} = \mathbf{C}_{\mathbf{lnK}_{i,t-1}\mathbf{H}_{i,t}}(\mathbf{C}_{\mathbf{H}_{i,t}\mathbf{H}_{i,t}} + \mathbf{R}_{i,t})^{-1} \tag{24}$$

where $e_{i,t}$ is the observation error, which is with zero mean and covariance $\mathbf{R}_{i,t}$.

4. Go back to step 2 and repeat the processes until all the observed heads are assimilated.



## 6.2 EnKF: Benchmark selection

We use only Scenario S4, i.e. the transient scenario, because the EnKF in its original form is a data assimilation framework. That means it works on time series of data in transient systems by design. There exist other versions of EnKFs that were extended to work, e.g. with iteration, on steady-state problems, but these other versions are not the focus of the current benchmarking example.

## 6.3 EnKF: Simulation validation

Figure 9 shows the mean and the standard deviation of the updated $\ln K$ realizations obtained by the reference solution (top row) and by the EnKF (bottom row) in the featured (transient) benchmarking scenario S4. In comparison with the pCN-PT reference solutions in the top row, we find that the EnKF successfully captures the main features of the reference solution in the mean (left column); however, it visually lacks some contrast. Also, the EnKF seems to specify a larger standard deviation (right column)

by visual comparison. Yet, the general shape of standard deviation, with reduced uncertainty in regions of the highest data density, is plausible.

## 6.4 EnKF: Metrics for accuracy

In this section, we compare the results obtained by the EnKF with the reference results obtained by pCN-PT based on the proposed metrics for accuracy in Section 3.4. The ensemble size of the EnKF is set as

1000. In Table 5, we provide the metrics of the comparison to the reference results in their original form



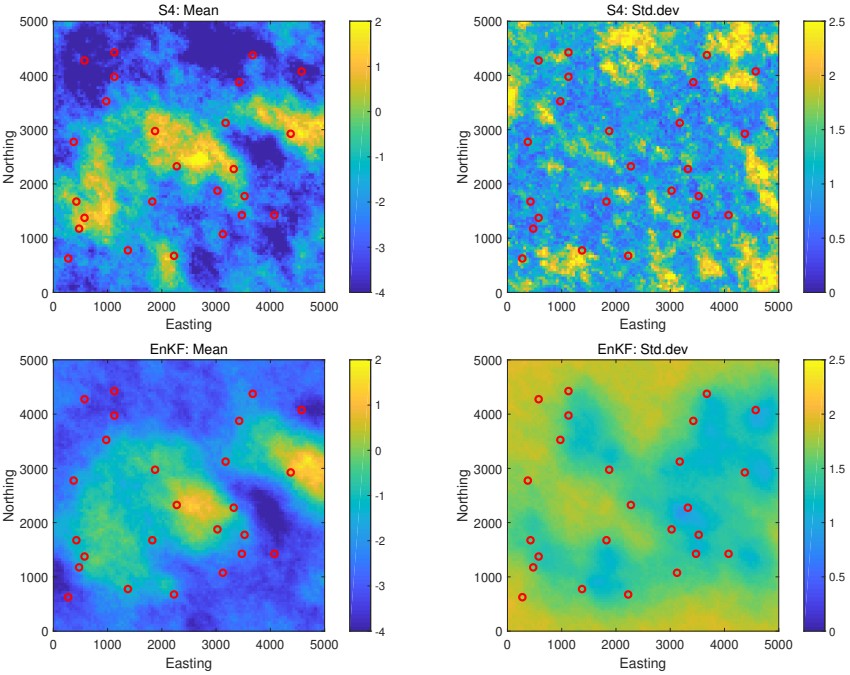

**Figure 9.** The mean (left column) and standard deviation (right column) of $\ln K$ realizations according to the reference solution (top row) and obtained by the EnKF (bottom row, $N = 1.000$ realizations) after the 10th (i.e. last) assimilation step in the featured (transient) benchmarking scenario S4.

(before normalization), after normalization, and transformed, according to what applies to each metric.

The shown metric values are close to each other and generally between 0.3 and 0.4. This indicates that

the EnKF (with ensemble size = 1000) has some deviation compared to the reference results. In addition,

it also shows that the accuracy of the EnKF (ensemble size = 1000) is consistent across different metrics.

To demonstrate the impact of a potentially worse result on the metrics, we computed the metric values

for EnKF using an ensemble size of 100. This is anticipated to yield higher values (lower accuracy) for

all metrics. The obtained metric values are shown in Table 6. It can be seen that, with a smaller ensemble

size, the metric values related to standard deviation (MAE std, RMSE std) become smaller while other





**Table 5.** Metrics values (original, normalized and transformed) of different versions for the EnKF with ensemble size 1000.

|                 | Original | Normalized | Transformed |
| --------------- | -------- | ---------- | ----------- |
| MAE mean        | 0.808    | 0.659      | 0.397       |
| MAE std         | 0.520    | 0.625      | 0.385       |
| RMSE mean       | 1.035    | 0.660      | 0.398       |
| RMSE std        | 0.605    | 0.657      | 0.396       |
| KS distance     | 0.341    | (-)[a]     | (-)[b]      |
| Energy distance | 8.073    | 0.379      | (-)[c]      |

[a] As the $KS$ distance is automatically normalized, we provide no additional normalization.

[b] As per definition $KS \in [0, 1]$, there is no need for additional transformation.

[c] Upon normalization, the Energy distance is automatically $\in [0, 1]$, so there is no need for additional transformation.

**Table 6.** Metrics values (original, normalized and transformed) of different versions for EnKF with ensemble size 100.

|                 | Original | Normalized | Transformed |
| --------------- | -------- | ---------- | ----------- |
| MAE mean        | 1.294    | 1.056      | 0.514       |
| MAE std         | 0.375    | 0.450      | 0.310       |
| RMSE mean       | 1.626    | 1.038      | 0.509       |
| RMSE std        | 0.480    | 0.521      | 0.343       |
| KS distance     | 0.477    | (-)        | (-)         |
| Energy distance | 11.893   | 0.545      | (-)         |

metric values become larger. Since EnKF is an MC-based approach, a smaller ensemble size may not

fully represent the distribution and can lead to biased estimation. A look at the updated standard deviation

map of $lnK$ (not shown here) reveals problems of filter collapse, i.e. the posterior standard deviation has

shrunk to become too small, which also makes the EnKF unable to continue doing productive updates in

later assimilation steps.





### 6.5 EnKF: Computing effort

As mentioned in Section 6.6, the compared numerical methods are evaluated in their accuracy and at

what computational price the accuracy is obtained. Whereas the posterior distributions were evaluated as

metrics, the computing costs are not compared, but simply reported as key performance indicators (KPIs)

and allow us to compare the methods among each other.

1. For the EnKF-1000 we have: 10 Time segments each with the 1000 samples result in $10 \cdot 1000$ forward

calls when counting formally; but the ten segments in a row come at the total cost of one full forward-

model run. Therefore, we count $N_f = 1000$. The computational effort of step 1 (initialization) and

step 3 (assimilation) is sometimes not negligible but does not require forward calls; thus they are

only accounted for in the wall clock time. With $N_f = 1000$ we get $LN_f = 3.0$ and $LN_f^* = 0.75$.

The overhead wall-clock time on a Intel(R) Core(TM) i7-7700 CPU @ 3.60GHz (4 cores) is $T_o =$

$\frac{211.129s - 115.85s}{211.129s} \approx 0.451$.

2. For the EnKF-100 we have: 10 Time segments each with the 100 samples result in cost equivalents of

$N_f = 100$ forward calls – and again the computational effort of steps 1 and step 3 are only accounted

for in the time measurement. With $N_f = 100$ we get $LN_f = 2.0$ and $LN_f^* = \frac{2}{3}$.

The overhead wall-clock time is on the same computer $T_o = \frac{66.102s - 55.017s}{66.102s} \approx 0.168$.





In both cases, the calculation of step 1 was not incorporated into the calculation as this was pre-

calculated. Parallelization was not tested; however, this could speed up the calculations considerably

(Houtekamer et al., 2014; Kurtz et al., 2016).

### 6.6   EnKF: Metric summary

In Figure 10, we summarize the accuracy and computational effort. Note that we only plot normalized

metrics on the right y-axis because normalization is often metric-dependent.

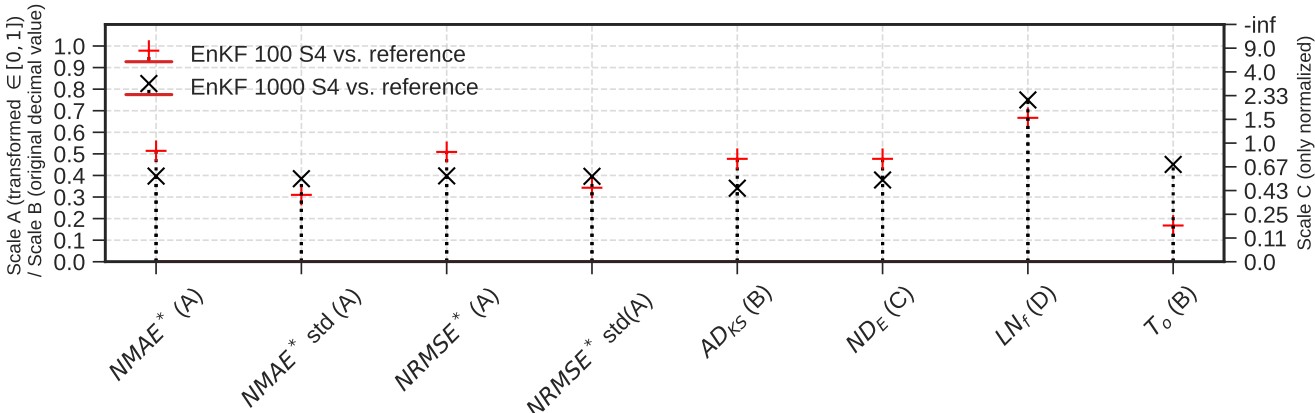

**Figure 10.** Comparison of the accuracy and computational effort of the EnKF with 100 and 1000 ensemble sizes to the reference results.
The correct axis for each metric is given in round brackets; Scale A is the transformed value which is the value normalized and then non-
linear squeezed into $[0, 1]$. Scale B corresponds to original values that were not normalized or transformed and Scale C corresponds to only
normalized metrics.

### 6.7   Discussion: performance of the EnKF evaluated through the Benchmarking metrics

The example showed nicely that the EnKF using 1000 ensemble members had overall a higher accuracy

than the EnKF with only 100 ensemble members, based on a comparison with the reference solution

generated by pCN-PT. The comparison also showed that this was at the expense of increased computing

costs.

Two of the plotted metrics showed counter-intuitive results because the posterior uncertainty (after data

assimilation) would be characterized slightly better by the 100-member EnKF than the 1000-member

EnKF. The posterior standard deviation for EnKF with 100 ensemble members is closer to the "true"

reference standard deviation by pCN-PT than EnKF with 1000 ensemble members. This can be attributed

to an overestimation of the posterior standard deviation by the EnFK-1000, which is countered by an

underestimation of the posterior standard deviation by EnKF-100 due to filter collapse. Therefore, by

lucky cancellation of errors, this brings the EnKF with 100 realizations closer to the average standard

deviation of the reference results, which causes smaller error metric values associated with the standard

deviation. The price to pay is an increase in the error metrics addressing the mean.

Therefore, it is advantageous and important to evaluate the performance of a method taking into account

all metrics, and for MAE and RMSE it can be noticed (see Table 6) that the mean shows much larger

deviations from the reference solution than the standard deviation, pointing to the underestimation of

the standard deviation by the small EnKF ensemble. We conclude that the metrics perform well for the

evaluation of a candidate method, but that the metrics should be evaluated carefully in the context of each

other and that their interpretation, supported by the plotted results, has to be done thoroughly.

## 7 Discussion and Conclusion

In this study, we have designed and implemented a suite of five well-defined benchmarking scenarios for groundwater flow with different levels of spatial heterogeneity, different spatial configurations of hydraulic head observations, different levels of data noise variance, and with groundwater flow at steady or transient state. We also developed highly accurate solutions for these five inversion benchmarking scenarios with the pCN-PT algorithm, which can serve as reference solutions for the community in future comparison studies.

We proposed a group of metrics to compare inverse modeling solutions with the reference solutions: metrics for accuracy of solution (MAE, RMSE, Kolmogorov-Smirnov distance, Energy distance, potential scale reduction factor), metrics for computing costs (number of forward calls, percentage overhead wallclock time, scaling efficiency) and metrics for ease of implementation and reproducibility of results.

To demonstrate and illustrate the intended application of these benchmarks, reference solutions, and metrics, we provided an example application of benchmarking at the example of a plain-vanilla EnKF (with 1000 and 100 ensemble members). The metrics that we proposed for the evaluation of candidate inverse methods performed well, with worse performance for an EnKF with a small ensemble size compared to an EnKF with a large ensemble size. However, the example also showed that it is important to evaluate the different metrics jointly and add visual comparisons of the results.

Reflecting on our reference solutions, we found in this work that the pCN-PT needs more sampling iterations to converge if more observations are available, but a better estimation of multi-Gaussian parameters can be achieved. Convergence of the pCN-PT is also slower with smaller error variance of observations and in case of stronger heterogeneity of parameter fields. In all those cases, the slower convergence is associated with more accurate results. We found for the transient case, that already a lot of computing power is needed to calculate a reference inverse solution with pCN-PT.

Reflecting back on our suggested benchmarking initiative, a logical endeavor is an extension to more realistic and more challenging cases. This would include Non-Multi-Gaussian parameter fields, which are often more realistic for subsurface media than Multi-Gaussian parameter fields given the presence of fractures and fluvial deposits. It would also include the extension to flow in unsaturated porous media (or multiphase flow) and/or coupled flow-transport processes. However, for all of these extensions, it would be much more challenging to derive a reference inverse solution. It was shown that pCN-PT can also handle Non-Multi-Gaussian parameter fields, but it will be difficult to achieve convergence for a transient flow case and a similar problem size as in the current study. Either a brute-force computing effort has to be undertaken to calculate reference solutions for such cases, or the pCN-PT algorithm has to be further optimized to become more efficient. As already quite a body of literature exists on solutions for Non-Multi-Gaussian flow problems, it can be expected that reference solutions for such cases are received with interest by the inverse modeling community. A reference inverse solution for a coupled 3D unsaturated-saturated flow problem will be even more challenging, given the non-linearity of the governing equation.



The challenge is then not limited to the more difficult sampling of the parameter space, but also to the much more expensive forward model run calls. As an alternative, we think that it could be of interest for the community to calculate reference inverse solutions with pCN-PT for 1D soil hydrological problems considering the uncertainty of all soil hydraulic parameters.

In addition to the above outcomes, this study aims to initiate a comparison study within the scientific community based on the suite of benchmarking scenarios for stochastic inverse modeling and reference solutions for the benchmarking scenarios. The (groundwater) inverse modeling community must work with benchmarks to compare (new) inverse modeling methods more easily with existing methods. In addition, estimates of posterior uncertainty can be better evaluated with the benchmark solutions. The algorithms, reference cases, and datasets are freely available to the community. We very much hope that they will be used by the community, and extended with cases including Non-Multi-Gaussian groundwater flow, unsaturated flow, and coupled flow and transport.

**Appendix A: EnKF results for the steady-state scenario S0**

Here, we show the results for scenario S0 with a steady-state flow problem and application of the EnKF. Figure A1 displays the mean, standard deviation, and K-S distance for this scenario and the application of the EnKF. We find that the updates are not able to reproduce the main features of the reference and have a large uncertainty and K-S distance.





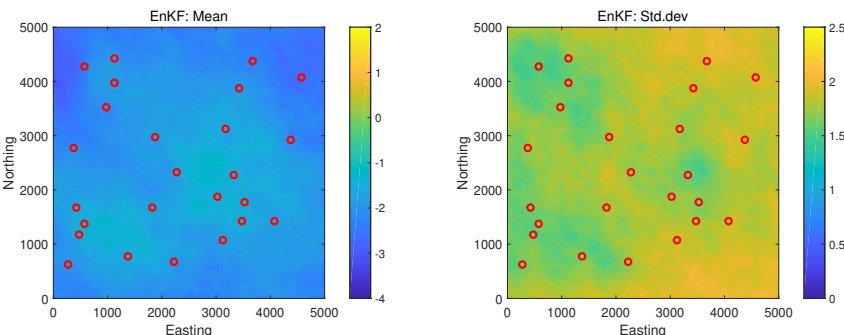

**Figure A1.** The mean (left upper column) and the standard deviation (right upper column) of updated lnK realizations for scenario S0 (steady-state groundwater flow problem).

**Appendix B: Details on the pCN-PT MCMC algorithm used for computing the reference solutions**

The pCN-PT proposed by Xu et al. (2020) is an efficient method for Bayesian Inversion of multi-Gaussian parameters. It combines the parallel tempering technique (Swendsen and Wang, 1986) with the pCN-MCMC (Cotter et al., 2013b). It employs several parallel Markov chains to sample from a series of tempered posterior distributions, making the target Chain (also called the cold chain) efficiently explore the target distribution after swapping with hot chains. Due to the pCN, it proposes samples that automatically

honor the prior multi-Gaussian distribution.

Briefly, the algorithm of the pCN-PT mainly consists of the following four steps:

1. Initialization step. An initial conductivity field $u_i^0$ for chain $i, i = 1, ..., n$ is generated following the multi-Gaussian distribution $u_i^0 \sim N(0, C)$, where $n$ is the total number of chains, and $C$ is the co-variance matrix of the random field. In addition, since the pCN-PT is a combination of the parallel

tempering technique and the pCN-MCMC, a temperature ladder $T_1 < T_2 < ... < T_i < ... < T_n$, with





$T_1 = 1$, a jumping factor ladder $\beta_1 < \beta_2 < ... < \beta_i < ... < \beta_n$, with $\beta_n < 1$, and a swap proposal frequency $d$ need to be designed. Here, the likelihood $L(u)$ of the hotter chains gets flattened by the temperature, inducing the posterior $\pi_t(u)$ to get flattened towards the prior as well. Therefore, the posterior $\pi_t(u)$ can be rewritten as below according to the Bayesian theorem:

$$\pi_t(u) \propto L(u)^{\frac{1}{T}} p(u) \tag{B1}$$

2. Proposal step. At the $k^{th}$ sampling iteration, a pCN proposal $v_i^k$ for all chains $i = 1, ..., n$ is calculated according to the proposal function, which is dependent on the sample $u_i^k$, the jumping factor $\beta_i$ and colored noise $\varepsilon$ (following the same distribution as the prior).

$$v_i^k = \sqrt{1 - \beta_i^2} u_i^k + \beta_i \varepsilon_i^k, \varepsilon_i^k \sim N(0, C) \tag{B2}$$

3. Acceptance step. Recall that the acceptance probability $a(u_i, v_i)$ of the traditional MCMC is dependent on the ratio of the likelihood $L(v_i)$, prior $p(v_i)$, and proposal density $q(v_i, u_i)$ of the proposed sample $v_i$ to those $L(u_i)$, $p(u_i)$, $q(u_i, v_i)$ of the previous sample $u_i$ (see Eq.(B3)) (Xu et al., 2020). Specifically for the pCN-PT, due to the assumption of a multi-Gaussian prior, $p(v_i) * q(v_i, u_i)$ equals $p(u_i) * q(u_i, v_i)$, and the likelihood $L(u_i)$ is flatted by the temperature $T_i$. Therefore, for each chain $i$ at the $k^{th}$ sampling iteration, Eq.(B3) for the acceptance probability can be rewritten as acceptance-pCN-MCMC Eq.(B4):





$$a(u_i, v_i) = min\left\{1, \frac{L(v_i) * p(v_i) * q(v_i, u_i)}{L(u_i) * p(u_i) * q(u_i, v_i)}\right\}$$ (B3)

$$a(u_i^k, v_i^k) = min\left\{1, \left[\frac{L(v_i^{k|y})}{L(u_i^{k|y})}\right]^{\frac{1}{T_i}}\right\}$$ (B4)

where $y$ are measured observations. Note that, when all observation errors and modeling errors follow Gaussian distributions, the log-likelihood $\phi(u) = lnL(u \mid y)$ can be given as:

$$\phi(u) = lnL(u \mid y) = ln\left\{(2\pi)^{-\frac{m}{2}} \parallel C_y \parallel^{-\frac{1}{2}} exp\left[-\frac{1}{2}(y - y^o)^T C_y^{-1}(y - y^o)\right]\right\}$$ (B5)

with

$$y = g(u) + \eta, \eta \sim N(0, C_n)$$ (B6)

where $y^o$ are simulated observations, $g(\cdot)$ denotes steady or transient groundwater flow model, $\eta$ is the measurement-and-model error, and $C_n$ is the assumed covariance matrix of these errors. Thus, Eq.(B4) can further be rewritten as:





$$a(u_i^k, v_i^k) = min\left\{1, exp\left[\frac{\phi(v_i^k) - \phi(u_i^k)}{T_i}\right]\right\} \tag{B7}$$

After that, accept $v_i^k$ with the acceptance probability $a(u_i^k, v_i^k)$, then $u_i^{k+1} = v_i^k$; otherwise, reject $v_i^k$,

then $u_i^{k+1} = u_i^k$

4. Swapping step. For any pairs of chains $i$ and $j$, swap values between them $u_i^k \rightleftharpoons u_j^k$ with swap

acceptance probability $a_s(u_i^k, u_j^k)$ according to the swap proposal frequency $d$ (if $\frac{k}{d} = integer$),

$$a_s(u_i^k, u_j^k) = min\left\{1, exp\left[(\phi(u_j^k) - \phi(u_i^k)) * \left(\frac{1}{T_i} - \frac{1}{T_j}\right)\right]\right\} \tag{B8}$$

*Code and data availability.* The reference data and our replication data are published and publicly available at https://doi.org/10.18419/darus-2382. The benchmarking codes are published at https://github.com/LS3-university-of-stuttgart/hydrological-inversion-benchmarking.

*Author contributions.* T. Xu drafted the original manuscript, while T. Xu and S. Reuschen collaborated to design and write the pCN-PT code under the guidance of N. Wolfgang. S. Xiao and N. Wildt provided assistance in organizing the data and codes. The paper was thoroughly reviewed and edited by all co-authors.





*Competing interests.* At least one of the (co-)authors is a member of the editorial board of Hydrology and Earth System Sciences.

*Acknowledgements.* Financial support to carry out this work was received from the Deutsche Forschungsgemeinschaft (DFG, German Re-

search Foundation) through COMPUFLOW-Projects 359880532 and Germany's Excellence Strategy-EXC-2075- 390740016. Teng Xu ac-

knowledges the financial support from the National Natural Science Foundation of China (Grant No. 42377046).





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
