# Peer review of "Towards a community-wide effort for benchmarking in subsurface hydrological inversion: benchmarking cases, high-fidelity reference solutions, procedure and a first comparison"

_Hydrology and Earth System Sciences, 2024_

## Referee Comment (RC2)

**Comments**

Peter K. Kitanidis
Stanford University
Stanford, CA 94305, USA
peterk@stanford.edu

May 7, 2024

**1 General Comments**

The authors are to be commended for their efforts to provide a platform to evaluate and compare inverse methods in Hydrogeology. A challenging job, to say the least. I liked the paper, which I found interesting and stimulating, but my comments will emphasize my concerns or points I would like to see better clarified.

It seems that the authors' approach is to provide some "benchmarking scenarios" then evaluate methods by comparing them with reference solutions. The term reference solution is first mentioned without explanation in the abstract. My first impression was that a reference solution is the ground truth, meaning the true or correct answer. It was later made clear that the term referred to a presumably best possible solution. (That's my interpretation. It would help immensely if the authors could explain the meaning right at the beginning.) However, one may wonder whether the reference solutions are best. Even 20000 samples from the posterior, which could be correlated, may not completely explore the probability space of the highly multivariate distribution. As a consequence, I find that some of the comparison metrics may be an overkill.

I am concerned that the methodology will benchmark only part of the solution of an inverse problem, which is much more than the algorithm to apply Bayes' theorem. I will explain what I mean in the next three paragraphs.

For me, an "inverse problem" is a problem in which the forward map and the data do not suffice to give a unique answer. Thus, using prior information (or regularization, structural information, or whatever else one may call it) is essential. I applaud the authors' emphasis on the Bayesian approach. Inverse modeling is a data science problem with all the consequences.

The first consequence is that inverse modeling is an iterative process in which data, other information, and the modeling objectives are considered; an approach is proposed, hyperparameters estimated, and the overall method is tested; then parts or all are modified; and so on, until convergence. This aspect

is seldom discussed in published papers, which promote the illusion of a one-way (noniterative) workflow: Put the data, then the forward map, run, and get results. The idea of having a one-way (noniterative) workflow is appealing but dangerous. How can one know what formulation to use unless one tries several? In the approach proposed in this paper, this aspect of inverse modeling is not considered or benchmarked.

Bayesian methods are very powerful but also tricky. There are reasons why a great scientist like R. A. Fisher was a lifelong critic. Bayesian methods have come a long way since Fisher criticized them, though not every scientist or engineer who applies them is aware of the advances. Many still think of Bayesian inference as one where Bayes theorem is used and nothing else. Furthermore, that prior information is "subjective" and "preordained" or somehow given, while the accuracy with which to reproduce the data is known beforehand. These are fallacies that I mention because they make the task of evaluating and benchmarking inverse-problem methods so much more challenging. My concern is that this paper is not helpful in dispelling these common misconceptions.

Turning my attention to benchmarking metrics, one general comment is that they are numerous, but all are what I call "point-centric" or "pixel-centic". In other words, they focus on the accuracy or errors at the smallest discretization scale. Consider the following:

1. The quantities we deal with in Hydrogeology, like the log-conductivity, have support volumes. It is understood that different models and applications may require parameters with different support volumes.

2. Computing the estimation variance at the finest scale is fraught with difficulties as it depends solely on the behavior near the origin of the chosen covariance function. For example, consider the difference between an exponential versus a Gaussian covariance. They result in very different computed variances of point estimates. Because small-scale variability is in the nullspace of the forward map, the assumed covariance dominates, while the assumed covariance has much less effect on computing large-scale estimates and variances of estimation.

3. In my experience, practitioners are not interested in the complete a posteriori pdf of, say, the log-conductivity because it is understood that not only is it hard to compute, but it also relies on many assumptions whose usefulness can go only so far. Instead, the question they ask is "What scales of variability are resolved?"

In my view, when it comes to inverse problems, one must consider "scale-centric" accuracy benchmarks, such as the accuracy of discrete cosine transform (DCT) coefficients. The DCT is an excellent tool for evaluating what scales are resolved. For example, we can evaluate whether the correct average is computed or variability at scales larger than 100 meter is resolved.

It is good to find a role for conditional simulations, meaning samples from the *a posteriori* distribution. I increasingly find conditional realizations, which

reveal the scales at which there is uncertainty, much more useful than MSE or confidence intervals.

My final comments are related to using the Gaussian distribution to generate log-conductivity fields and data. Some may consider that this invalidates the results because the model used in the inversion is the same as the "true model". I am coming to this issue from a different angle. The Gaussian model cleverly deployed has been successful in Hydrogeology, River Bathymetry, and Face Recognition, but this does not mean that the true unknown is somehow Gaussian. The strength of Gaussian models (with variable transformations) is their versatility and robustness, regardless of the unknowns. I remember that George E. P. Box's, a pioneer of modern Bayesianism, gave us the memorable aphorism that "All models are wrong, but some are useful." I teach my students that modeling assumptions may be appropriate and useful *up to a point*.

Stochastic methods in inverse modeling can easily get out of hand and ignore the basic premise that inverse modeling means *estimation with limited information* and consequently should have limited objectives. For example, we can determine a mean value or a variance of certain unknown components, but we cannot estimate everything. Just because one can compute something does not suggest that one should! And just because a modeling assumption is useful for some purposes does not mean it is useful for all purposes! I question estimating something that relies critically on assumptions that cannot be verified, such as the complete distribution at the smallest scale.

The preoccupation with complete distributions is a legacy of classical (frequentist) methods and has no place in inverse modeling with Bayesian methods. In Bayesian methods, the unknowns have distributions that represent a state of knowledge, not the true distribution of the unknown.

With these thoughts in mind, I find criteria such as the K-S distance an overkill. Also, I would suggest including cases not generated by Gaussian distributions.

**2  Specific Comments**

I take issue with the statement that the objective is to learn about the parameters "by matching" (line 142). I would rephrase with the perhaps loftier "by assimilating information in the hard and other data (or the prior)". Too much emphasis on data matching is wrong. Many parameter sets can match data, yet they can be poor estimates. One of the key questions in solving an inverse problem is how closely to match the data.

Add "and uniform variance" (line 161).

I am unsure if the readers understand "best estimate and its variance" (line 174) since they have not been defined.

I am not excited about the normalization (line 184). The no-error case is defined as full agreement with the presented reference case. However, both the candidate and the reference solution must have some error, particularly at the point scale. The criterion will be too dependent on the small-scale errors that

are of limited interest and too dependent on assumptions.

Specifically for estimating log-conductivity, mean square and mean absolute errors are dimensionless numbers that, in my view, hardly need normalization.

I would suggest "hydraulic head or pressure, as appropriate" (Line 226).

The normalization through a sigmoid (Eq. 3) is interesting. I have no experience with it, but I suspect it will result in bunching together the bad solutions near the highest value, 1.

Is there a mathematical guarantee that the metrics of Eq. 5 or 8 will not exceed 1?

It remains to be seen whether the K-S distance at each node is useful. My first reaction is that it is an overkill. I am unsure how this will be applied and, most importantly, is it really important?

Regarding Eq. 12, is the symbol E supposed to mean numerical average? Also, I do not know what $D_E$ signifies.

Regarding the number of forward-map calls and using Eq. 3, expensive solutions will bunch together. For example, for 10000 calls that take a day, the normalized value is 0.8. For 1000000 calls that take 100 days, the normalized value is 0.8571. The normalized-value difference seems small, while the actual difference between 1 and 100 days of computations is consequential.

In my experience, using wall clock time (or CPU time, for that matter) is pointless. It depends so much on the computer system and how busy it is with other jobs.

One important issue regarding evaluating computational effort is related to the iterative nature of most methods. One can game the effort metric by using a "starting" solution that is cleverly selected close to the final, thus reducing the required iterations.

The value -2.5 for the mean (Line 400) implies some units for conductivity (like meters per second). Please clarify. Same for $\sigma_e$ (Line 409).

By observation error standard deviation, I assume it is to be used to introduce errors in generated data. However, I assume it is not to be used in solving the inverse problem. It is a pity that this information is given; in more realistic cases, deciding what to use is one of the most important steps in a method.

The sentence in Lines 433-436 is unclear.

Regarding the "thinning out" of samples in the chain, I would have expected that the main reason would be to have independent samples. Are the 20000 samples independent? I have enough experience with MCMC methods to know that they are tricky. What assurances are there that the solution is a "high-quality" solution?

On Line 502, I suggest "the better the identification that can be achieved."

**3   Technical Corrections**

No comments here.

I would like to take the opportunity to thank the authors for their work.

---

## Referee Comment (RC3)

Review of "Towards a community-wide effort for benchmarking in subsurface hydrological inversion: benchmarking cases, high-fidelity reference solutions, procedure and first comparison"

**General comments**

The paper presents a very interesting topic and can be useful for the community. I think it needs some clarification in the writing as it is difficult to follow the full discussion. There is the need to provide some clarification for when the authors reference to the truth case, the reference and the benchmark scenarios. Consistency on how the different conditions are referred to will help the reader.

One question is about the added information that would come from the two synthetic truths: they only have a different standard deviation and I am not sure there is much to gain from comparing the two of them. I wonder if it would be more useful to compare different hydrogeological conditions. Something like bookend scenarios that look at very different assumptions would be very beneficial for the reader to understand the power of the methods presented.

Looking at the scenarios in table 3, the regular well distribution scenario does not seem to add more information and S0, S2 and S3 do not really represent a variety of conditions.

My comments below are mostly about Figures: the paper would really benefit from a more consistent representation and discussion of the results and more consistency in the figures with results.

I would also suggest to provide some more informative conclusions, maybe summarized in a bullet point lists.

**Specific comments**

Line: suggest to change to "across multiple scales"

Line 37: I would not say that data scarcity and subsurface heterogeneity are the sole responsible for uncertainty. Depending on the model, we need to add uncertainty on other terms of the water budget.

Line 413: why S1 is a fallback scenario? I wonder if S1 is needed, and if yes I think there is the need to have at least to scenarios with regular well distribution to be able to make a comparison

Figure 2: as mentioned earlier: is it a limiting factor having two truths that are so similar and only different for standard deviation?

Figure 3: I think that the use of a different scale for S2 is misleading, but if the authors believe that it is important I would suggest to add explanation

Figure 4: same comment on the scale

Figure 5: same comment: it would be good to add an explanation

Line 514: it is really hard to see the sharp decrease

Figure 6: the legend for S0 says "reference" and for the other scenarios "Synthetic truth". Is this correct? Also, are there some conclusion to be derived from the mean always being above the reference/synthetic truth?

Figure 9: is the reference solution, the true solution?

**Technical corrections**

No technical corrections.

---

## Author Response (AR1)

**COMMENTS FROM EDITOR**

**Editor decision: Reconsider after major revisions (further review by editor and referees) by Mauro Giudici**

The reviewers provided very detailed reports: in some cases, their arguments are very accurate and far-reaching. I think the authors should profit from this opportunity to deeply revise the manuscript, in order to focus several aspects in a better way.

I notice that two of the reviewers raised concerns about the use of random fields, which share the same statistical structure, with different ranges of K values, but without a strong variation in the hydrogeological structure between the two test cases. I recommend the authors to deeply reconsider this question, which - by the way - somehow interacts with the comments about scaling from reviewer #2. The answer proposed in the reply to reviewer #3's comments ("Furthermore, it would be intriguing to undertake a comparative analysis of various hydrogeological conditions to gain a comprehensive understanding of the efficacy of inverse methods.") is too generic and not fully satisfactory, in my opinion.

*Thank you for your comments and suggestions.*

*We are grateful for the valuable comments and suggestions provided by the editor and reviewers, which have greatly contributed to the revision and improvement of our manuscript.*

*We have not shown a strong variation in the hydrogeological structure, such as a comparative scenario with non-Gaussianity, due to the fact that EnKF and pCN-PT employed in this study are specifically designed to handle Gaussian problems. In our future work, we will explore non-Gaussian scenarios and compare different inverse modeling methods suited for non-Gaussian problems. This will not only require different benchmarking scenarios, but also different reference algorithms, and will attract a different type of inversion candidate algorithms for benchmarking.*

*Regarding the comments on scaling from reviewer #2, we maintain that this is beyond the scope of this work and acknowledge this limitation in the manuscript in lines 700-705:*

> *"It would also be interesting to evaluate the resolved scales of variability. The evaluation of inverse methods that are applied to the benchmarking cases could also consider the reproduction of scale-dependent variability, using for example scale centric accuracy benchmarks such as discrete cosine transform coefficients. It would be helpful then that benchmark solutions are calculated at a high spatial discretization and for more complex subsurface spatial patterns but this was beyond the scope of this work, but certainly an interesting future endeavor."*

*In response to reviewer #3's comments ("Furthermore, it would be intriguing to undertake a comparative analysis of various hydrogeological conditions to gain a comprehensive understanding of the efficacy of inverse methods."), it is not realistic to compare the impact of all hydrogeological conditions in a single paper due to their diversity. However, we welcome the community to explore alternative hydrogeological conditions in future research endeavors.*

**Editor initial decision: Start review and discussion after technical corrections by** Mauro Giudici

The submitted manuscript suggests the use of two K-fields, sharing the same correlation structure, but with different variances, to define different scenarios for benchmarking stochastic methods of inversion. In my opinion, every choice of benchmark scenarios could be criticized, but I appreciate the effort of providing a clear testing environment. Therefore, the work could rise some objections from reviewers, but I think it can enter the discussion stage. I list below some comments, that could be considered by the authors before the start of the discussion stage.

*Thank you for your comments and suggestions on our manuscript. Comments and suggestions are all valuable and very helpful in revising and improving our paper. We have carefully considered the comments and have made corrections that we hope will meet with your approval.*

Why only two K fields are considered? In particular, they share the same correlation structure, so that they do not represent a wide variety of hydrogeological conditions.

*Thank you for your comment. Although the two K fields share the same correlation, they have different variances: one with a standard deviation of 2 and the other with a standard deviation of 1. There are many hyperparameters used for generating the K fields, and it is not realistic to compare the effect of all hyperparameters in one paper. However, we welcome the community to test other hyperparameters in the future.*

Line 140. I would substitute "geostatistics" with "numerical modeling". The word "geostatistical" is often used, but in many places I would rather use "stochastic".

*Corrected as suggested.*

Line 157. Neglecting the marginal likelihood is common practice, but I do not fully agree with this simplification.

*The marginal likelihood remains constant for any given (fixed) data set and a given combination of geostatistical prior and forward simulation code, so disregarding it will not impact the results and can help reduce computational burden.*

Lines 160 to 168. I think that the choice of a Gaussian distribution for errors and a log normal prior distribution could somehow constrain inversion results: I think that different alternatives could be useful for a more fair intercomparison. The work is extensively based on a paper published by some of the authors on WRR (DOI:10.1029/2020WR027110). Since that method is tailored to multi-Gaussian K

fields, it should be more clearly discussed the effects of this assumption on the expected results.

*Thanks for your suggestion. We chose to use multi-Gaussian distribution in our work because the EnKF, QLGA, Sequential Estimator, Pilot Points and pCN-PT are designed to handle Gaussian problems. In our future work, we will explore non-Gaussian scenarios and compare different inverse modeling methods suited for non-Gaussian problems. This will not only require different benchmarking scenarios, but also different reference algorithms, and will attract a different type of inversion candidate algorithms for benchmarking.*

Section 3.3. I think that it could be useful to include among the validation tools also the simulation of flow fields for conditions different from those used for calibration, e.g., when boundary conditions or position and rate of pumping wells change.

*We really appreciate your suggestion. It will be interesting to explore the impact of different boundary conditions and changes in the position and rate of pumping wells. However, this study is just a preliminary comparison. We acknowledge that there are many other factors that we have not considered in this work, and we look forward to engaging with the community to enrich our research in the future.*

Equations (20) and (21) refer to 3D flow, but are then used for 2D flow conditions, under the hydraulic assumption, for a horizontal, confined aquifer, with constant thickness. I think this should be explicitly mentioned.

*Thanks for your suggestion. We have added the following statement in line 388:*

> *"Although Eq.20 and Eq.21 are originally for 3D flow, in this work they are applied to 2D flow conditions."*

Figure 1. The figure caption should explain what is represented by the red square centered on #3 well.

*We appreciate your suggestion. The red square has been removed from Fig. 1. Thank you for bringing it to our attention.*

Figure 2. If I understood properly what is done, a single plot would be sufficient and the color scale could be differentiated for the two cases, simply by specifying the extreme values.

*Thank you for your suggestion. This figure displays two synthetic truths of lnK, each with a different standard deviation. We believe that showing these two plots would help readers better understand.*

**COMMENTS FROM Referee #1**

This is a much-needed work for the benchmarking of stochastic inversion techniques. My only comment would be that the choice of log multi-Gaussian random fields with a small variance makes the problem relatively simple, yet it could provide a way to benchmark different inverse methods.

Will the benchmarking data be published alongside this manuscript?

*Thank you so much for your valuable comments and recommendations on our manuscript. As depicted in Figure 2 and Table 4, we have designed two lnK fields with different variances: one with a variance of 4 and the other with a variance of 1. These two lnK fields are indeed meant to be used as benchmarks for different inverse methods. We believe that having a small variance is an important scenario where linearized and quasi-linear methods can also be compared, such as Successive linear estimator (SLE) by Yeh and others, the quasi-linear approach by Kitanidis, and implicitly linearized methods like the EnKF. Notice however that the scenario with a variance of 4 is not a scenario with a small variance, in our opinion.*

*Yes, the reference and benchmarking data, as well as the benchmarking codes, are (now) publicly available at https://doi.org/10.18419/darus-2382 and https://github.com/LS3-university-of-stuttgart/hydrological-inversion-benchmarking, as noted in the manuscript in lines 743-744.*

**COMMENTS FROM Prof. Peter K. Kitanidis**

**1. General Comments**

The authors are to be commended for their efforts to provide a platform to evaluate and compare inverse methods in Hydrogeology. A challenging job, to say the least. I liked the paper, which I found interesting and stimulating, but my comments will emphasize my concerns or points I would like to see better clarified.

It seems that the authors' approach is to provide some "benchmarking scenarios" then evaluate methods by comparing them with reference solutions. The term reference solution is first mentioned without explanation in the abstract. My first impression was that a reference solution is the ground truth, meaning the true or correct answer. It was later made clear that the term referred to a presumably best possible solution. (That's my interpretation. It would help immensely if the authors could explain the meaning right at the beginning.) However, one may wonder whether the reference solutions are best. Even 20000 samples from the posterior, which could be correlated, may not completely explore the probability space of the highly multivariate distribution. As a consequence, I find that some of the comparison metrics may be an overkill.

*Thank you so much for your valuable comments. The reference solutions are the best possible solutions with best estimate and posterior standard deviation (as mentioned in lines 443-444 of the original version of the manuscript). We have now included an explanation of the reference solutions in the abstract (lines 6-9).*

*"However, in past studies until now, comparisons were made among approximate methods without firm reference solutions. Note that the*

*reference solutions are the best possible solutions with best estimate, and posterior standard deviation and so forth."*

*The 20000 samples are thinned out from 800000 posterior samples. They could still be correlated. However, the Potential scale reduction factor (PSRF) and posterior trace inspection have shown good exploration of the posterior samples.*

I am concerned that the methodology will benchmark only part of the solution of an inverse problem, which is much more than the algorithm to apply Bayes' theorem. I will explain what I mean in the next three paragraphs.

For me, an "inverse problem" is a problem in which the forward map and the data do not suffice to give a unique answer. Thus, using prior information (or regularization, structural information, or whatever else one may call it) is essential. I applaud the authors' emphasis on the Bayesian approach. Inverse modeling is a data science problem with all the consequences.

The first consequence is that inverse modeling is an iterative process in which data, other information, and the modeling objectives are considered; an approach is proposed, hyperparameters estimated, and the overall method is tested; then parts or all are modified; and so on, until convergence. This aspect is seldom discussed in published papers, which promote the illusion of a one- way (noniterative) workflow: Put the data, then the forward map, run, and get results. The idea of having a one-way (noniterative) workflow is appealing but dangerous. How can one know what formulation to use unless one tries several? In the approach proposed in this paper, this aspect of inverse modeling is not considered or benchmarked.

*Yes, inverse modeling is an iterative process. We have duly considered and implemented the iterative process mentioned in our work, which has sparked numerous discussions. However, due to spatial constraints and the need to maintain topic coherence, it is challenging to comprehensively explore all facets in a single paper. Therefore, as stated in our paper (e.g., lines 125-129), "we cordially welcome the scientific community to apply these benchmarking scenarios and reference solutions and evaluate their candidate inverse modeling methods in a multi-objective manner that will fairly and transparently reveal trade-offs between computational intensity, achievable accuracy, (non-)intrusiveness to forward simulation codes, robustness against non-linearities and limits of applicability posed by more or less restrictive assumptions."*

*Notably, hyperparameter estimation has been thoroughly conducted and published in our previous work (Xiao et al., 2021), as also referenced in lines 487-489."*

*"The algorithm for pCN-PT is provided in the appendix, and full details can be found in Xu et al. (2020). Possible extensions for multi-facies aquifers with internal (Gaussian) heterogeneity or to cases with uncertain covariance parameters exist (e.g., Xiao et al., 2021)."*

*"Xiao, S., Xu, T., Reuschen, S., Nowak, W., and Hendricks Franssen, H.-J.:*

*Bayesian Inversion of Multi-Gaussian Log-Conductivity Fields With Uncertain Hyperparameters: An Extension of Preconditioned Crank-Nicolson Markov Chain Monte Carlo With Parallel Tempering, Water Resources Research, 57, e2021WR030 313, 2021."*

Bayesian methods are very powerful but also tricky. There are reasons why a great scientist like R. A. Fisher was a lifelong critic. Bayesian methods have come a long way since Fisher criticized them, though not every scientist or engineer who applies them is aware of the advances. Many still think of Bayesian inference as one where Bayes theorem is used and nothing else. Furthermore, that prior information is "subjective" and "preordained" or somehow given, while the accuracy with which to reproduce the data is known beforehand. These are fallacies that I mention because they make the task of evaluating and benchmarking inverse-problem methods so much more challenging. My concern is that this paper is not helpful in dispelling these common misconceptions.

*We agree. Prior information possesses a certain degree of subjectivity. The efficacy of the a priori would directly impact the accuracy and reliability of the estimation outcomes. This paper truly has not discussed this part, since it is beyond the scope of the paper. This is acknowledged now on lines 95-96:*

*"(prior information inherently contains a certain degree of subjectivity, and its effectiveness directly influences the accuracy and reliability of estimation outcomes)"*

Turning my attention to benchmarking metrics, one general comment is that they are numerous, but all are what I call "point-centric" or "pixel-centic". In other words, they focus on the accuracy or errors at the smallest discretization scale. Consider the following:

1. The quantities we deal with in Hydrogeology, like the log-conductivity, have support volumes. It is understood that different models and applications may require parameters with different support volumes.

2. Computing the estimation variance at the finest scale is fraught with difficulties as it depends solely on the behavior near the origin of the chosen covariance function. For example, consider the difference between an exponential versus a Gaussian covariance. They result in very different computed variances of point estimates. Because small-scale variability is in the nullspace of the forward map, the assumed covariance dominates, while the assumed covariance has much less effect on computing large-scale estimates and variances of estimation.

3. In my experience, practitioners are not interested in the complete a posteriori pdf of, say, the log-conductivity because it is understood that not only is it hard to compute, but it also relies on many assumptions whose usefulness can go only so far. Instead, the question they ask is "What scales of variability are resolved?"

In my view, when it comes to inverse problems, one must consider "scale- centric" accuracy benchmarks, such as the accuracy of discrete cosine transform (DCT)

coefficients. The DCT is an excellent tool for evaluating what scales are resolved. For example, we can evaluate whether the correct average is computed or variability at scales larger than 100 meter is resolved.

It is good to find a role for conditional simulations, meaning samples from the *a posteriori* distribution. I increasingly find conditional realizations, which reveal the scales at which there is uncertainty, much more useful than MSE or confidence intervals.

> *Thanks for these comments. We tend to agree with the notion that local estimates of posterior variance are very sensitive to the adopted covariance function and less used in practice as their estimates might be quite uncertain, for example, because systematic model errors are difficult to characterize and would contribute a lot to total uncertainty. We also agree with the fact that the resolved scales of variability are important for the practitioner. However, these points are mostly beyond the scope of this paper. Inverse methods that are applied on the benchmarking cases could also focus on the reproduction of scale dependent variability. It would be helpful then that benchmark solutions are calculated at a high spatial discretization but given the compute intensity of those calculations this is still very challenging. We acknowledge this limitation in the manuscript in lines 700-705:*

>> *"It would also be interesting to evaluate the resolved scales of variability. The evaluation of inverse methods that are applied to the benchmarking cases could also consider the reproduction of scale-dependent variability, using for example scale centric accuracy benchmarks such as discrete cosine transform coefficients. It would be helpful then that benchmark solutions are calculated at a high spatial discretization and for more complex subsurface spatial patterns but this was beyond the scope of this work, but certainly an interesting future endeavor."*

My final comments are related to using the Gaussian distribution to generate log-conductivity fields and data. Some may consider that this invalidates the results because the model used in the inversion is the same as the "true model". I am coming to this issue from a different angle. The Gaussian model cleverly deployed has been successful in Hydrogeology, River Bathymetry, and Face Recognition, but this does not mean that the true unknown is somehow Gaussian. The strength of Gaussian models (with variable transformations) is their versatility and robustness, regardless of the unknowns. I remember that George E. P. Box's, a pioneer of modern Bayesianism, gave us the memorable aphorism that "All models are wrong, but some are useful." I teach my students that modeling assumptions may be appropriate and useful up to a point.

Stochastic methods in inverse modeling can easily get out of hand and ignore the basic premise that inverse modeling means *estimation with limited information and consequently should have limited objectives*. For example, we can determine a mean value or a variance of certain unknown components, but we cannot estimate everything. Just because one can compute something does not suggest that one should! And just because a modeling assumption is useful for some purposes does not mean it is useful for all purposes! I question estimating something that relies critically

on assumptions that cannot be verified, such as the complete distribution at the smallest scale.

The preoccupation with complete distributions is a legacy of classical (frequentist) methods and has no place in inverse modeling with Bayesian methods. In Bayesian methods, the unknowns have distributions that represent a state of knowledge, not the true distribution of the unknown.

With these thoughts in mind, I find criteria such as the K-S distance an overkill. Also, I would suggest including cases not generated by Gaussian distributions.

*Thanks for your suggestion. We consider the K-S distance to be valuable, since KS distance at each node can give a more comprehensive comparison between the candidate and the reference. For a reference that may not be the "best", the averaged KS could provide a more meaningful comparison. As evidenced by the averaged K-S distances presented in Tables 5-6 for the EnKF with ensemble sizes of 1000 and 100, we observe that the K-S distance for the EnKF with an ensemble size of 1000 is smaller than that for the EnKF with an ensemble size of 100, indicating a closer resemblance to the reference.*

*We chose to use multi-Gaussian distribution in our work because the EnKF, QLGA, Sequential Estimator, Pilot Points Method and pCN-PT are designed to handle Gaussian problems. In our future work, we will explore non-Gaussian scenarios and compare different inverse modeling methods suited for non-Gaussian problems. This will not only require different benchmarking scenarios, but also different reference algorithms, and will attract a different type of inversion candidate algorithms for benchmarking.*

**2 Specific Comments**

I take issue with the statement that the objective is to learn about the parameters "by matching" (line 142). I would rephrase with the perhaps loftier "by assimilating information in the hard and other data (or the prior)". Too much emphasis on data matching is wrong. Many parameter sets can match data, yet they can be poor estimates. One of the key questions in solving an inverse problem is how closely to match the data.

*Rephrased as suggested in lines 147-148.*

*"The purpose of groundwater model inversion is to calibrate this model, i.e. to learn about $\vartheta$ by assimilating information in the hard and other data (or the prior)"*

Add "and uniform variance" (line 161).

*Added as suggested.*

I am unsure if the readers understand "best estimate and its variance" (line 174) since they have not been defined.

*"best estimate and its variance" correspond to the posterior mean and the posterior variance, respectively, which is rephrased in the paper in line 179.*

*"The posterior mean and its posterior variance are"*

I am not excited about the normalization (line 184). The no-error case is defined as full agreement with the presented reference case. However, both the candidate and the reference solution must have some error, particularly at the point scale. The criterion will be too dependent on the small-scale errors that are of limited interest and too dependent on assumptions.

Specifically for estimating log-conductivity, mean square and mean absolute errors are dimensionless numbers that, in my view, hardly need normalization.

*We acknowledge your suggestion; however, as stated in lines 189-190, our normalization process aims to achieve easily interpretable visual diagnostics across multiple metrics by ensuring that all metrics are scaled within the range of [0,1]. However, we agree that for some specific metrics or application environments where normalization may not be appropriate they should remain unnormalized. It has been emphasized in lines 191-192:*

*"although there are specific metrics or application environments where normalization may not be appropriate and should remain unnormalized."*

I would suggest "hydraulic head or pressure, as appropriate" (Line 226).

*Corrected as suggested.*

The normalization through a sigmoid (Eq. 3) is interesting. I have no experience with it, but I suspect it will result in bunching together the bad solutions near the highest value, 1.

*The nonlinear transformation is visualized below. For very bad solutions with $METRIC \gg 1$, the transformed metric $METRIC^*$ will be close to 1. This will result in bunching together very bad solutions. But in this case, it tells us these solutions are very bad (even much worse than the prior). And it doesn't make too much sense to have a clear comparison between very bad solutions.*

[Figure]

Is there a mathematical guarantee that the metrics of Eq. 5 or 8 will not exceed 1?

*Eq. 5 and 8 could exceed 1, which means the solution is worse than the prior. After the nonlinear transformation, it will be less than 1.*

It remains to be seen whether the K-S distance at each node is useful. My first reaction is that it is an overkill. I am unsure how this will be applied and, most importantly, is it really important?

*We consider the K-S distance to be valuable, since the KS distance at each node can give a more comprehensive comparison between the candidate and the reference. For a reference that may not be the "best", the averaged KS could provide a more meaningful comparison. As evidenced by the averaged K-S distances presented in Tables 5-6 for the EnKF with ensemble sizes of 1000 and 100, we observe that the K-S distance for the EnKF with an ensemble size of 1000 is smaller than that for the EnKF with an ensemble size of 100, indicating a closer resemblance to the reference.*

Regarding Eq. 12, is the symbol E supposed to mean numerical average?

*The symbol $E$ on the right-hand side of Eq. 12 denotes the mathematical expectation, whereas the subscript E in $D_E$ represents energy.*

Also, I do not know what DE signifies.

*As stated in line 294, $D_E$ denotes energy distance.*

Regarding the number of forward-map calls and using Eq. 3, expensive solutions will bunch together. For example, for 10000 calls that take a day, the normalized value is 0.8. For 1000000 calls that take 100 days, the normalized value is 0.8571. The normalized-value difference seems small, while the actual difference between 1 and 100 days of computations is consequential.

*Thanks for your comment. We agree with the comment. Though our normalization process aims to achieve easily interpretable visual diagnostics across multiple metrics by ensuring that all metrics are scaled within the range of [0,1], however, for some specific metrics or application environments where normalization may not be appropriate and should remain unnormalized. It has been emphasized in lines 191-192:*

> *"although there are specific metrics or application environments where normalization may not be appropriate and should remain unnormalized."*

In my experience, using wall clock time (or CPU time, for that matter) is pointless. It depends so much on the computer system and how busy it is with other jobs.

*We appreciate your suggestion. It is imperative to ensure that all scenarios are rigorously tested within a consistent computing environment, but we acknowledge some limitations of these metrics. This is acknowledged now on lines 624-626:*

> *"Note that we use this metric although it is evident that the consumed wall clock time also depends on some external poorly controllable factors not directly linked to the efficiency of the inverse modeling framework."*

One important issue regarding evaluating computational effort is related to the

iterative nature of most methods. One can game the effort metric by using a "starting" solution that is cleverly selected close to the final, thus reducing the required iterations.

*Thanks for your suggestion. This suggestion is highly valuable and we intend to incorporate it into our future research endeavors.*

The value -2.5 for the mean (Line 400) implies some units for conductivity (like meters per second). Please clarify. Same for $\sigma_e$ (Line 409).

*Please note that this study employs dimensional analysis without specific units, and any coherent set of units will produce equivalent results. We have added this statement in lines 390-391:*

*"Please note that this study employs dimensional analysis without specific units, and any coherent set of units will produce equivalent results."*

By observation error standard deviation, I assume it is to be used to introduce errors in generated data. However, I assume it is not to be used in solving the inverse problem. It is a pity that this information is given; in more realistic cases, deciding what to use is one of the most important steps in a method.

*We appreciate your suggestion. We will consider it in our future work and acknowledge this limitation in the manuscript in lines 448-451:*

*"Note that the measurement error is given here, but in reality, the measurement error is uncertain. Although often reliable information is available, for example considering instrument precision, the uncertainty of the measurement error could also be considered in an inverse modeling approach."*

The sentence in Lines 433-436 is unclear.

*The sentence should be: "Please be aware that the goal of benchmarking is not to be as close to the synthetic fields as possible (with a posterior uncertainty as small as possible), but to be as close to a high-end reference solution as possible (i.e. close in best estimate, posterior standard deviation and so forth)". We have read it and think it is clear enough.*

Regarding the "thinning out" of samples in the chain, I would have expected that the main reason would be to have independent samples. Are the 20000 samples independent? I have enough experience with MCMC methods to know that they are tricky. What assurances are there that the solution is a "high- quality" solution?

*The 20000 samples are thinned out from 800000 posterior samples. They could still be correlated. As we stated in lines 502-503, thinning out to 20,000 samples is for the memory saving purpose.*

*Considering the inherent nature of MCMC, each generated sample relies on the previous one. We have extensively evaluated various aspects to ensure a solution of high quality, such as achieving convergence in likelihood evolution and*

*obtaining mean and maximum potential scale reduction factor R values close to 1.2.*

 On Line 502, I suggest "the better the identification that can be achieved."

*Replaced as suggested.*

**3 Technical Corrections**

 No comments here.
 I would like to take the opportunity to thank the authors for their work.

*Thank you once again for your valuable comments and suggestions on our manuscript. Your feedback has been very helpful in revising and improving our paper. We have carefully considered all of the comments and made necessary corrections to ensure that they align with your expectations.*

**COMMENTS FROM Referee #3**

**General comments**

 The paper presents a very interesting topic and can be useful for the community. I think it needs some clarification in the writing as it is difficult to follow the full discussion. There is the need to provide some clarification for when the authors reference to the truth case, the reference and the benchmark scenarios. Consistency on how the different conditions are referred to will help the reader.

*Thanks for your suggestion. We have provided a detailed description of the benchmarking scenarios and synthetic cases in Sections 4.2 and 4.3, respectively. The clarity of the reference solution may be improved, and as such, we have included the following explanation of the reference solutions in the abstract (lines 6-9):*

*"However, in past studies until now, comparisons were made among approximate methods without firm reference solutions. Note that the reference solutions are the best possible solutions with best estimate, and posterior standard deviation and so forth."*

 One question is about the added information that would come from the two synthetic truths: they only have a different standard deviation and I am not sure there is much to gain from comparing the two of them. I wonder if it would be more useful to compare different hydrogeological conditions. Something like bookend scenarios that look at very different assumptions would be very beneficial for the reader to understand the power of the methods presented.

*Thank you for your comment. The rationale for utilizing two synthetic truths with varying standard deviations is to assess the efficacy of methods in handling fields with low and high heterogeneity.*

*We greatly value your suggestion and believe that it would be interesting to conduct a comparative analysis of diverse hydrogeological conditions. We will certainly take this into consideration and encourage the community to explore alternative hydrogeological conditions in future research endeavors. This has been added in the discussion and conclusion section in lines 705-707:*

> *"Furthermore, it would be intriguing to undertake a comparative analysis of various hydrogeological conditions to gain a comprehensive understanding of the efficacy of inverse methods."*

Looking at the scenarios in table 3, the regular well distribution scenario does not seem to add more information and S0, S2 and S3 do not really represent a variety of conditions.

*Thanks for your suggestion. As we mentioned in lines 419-421, Scenario S1 with the regular monitoring network is a fallback scenario for the comparison with irregular monitoring networks. The close spacing of some monitoring wells in the irregular networks may present challenges for certain methods due to their high autocorrelation. The standard deviations of lnK synthetic truths and measurement errors are different for S0, S2, and S3, and the impact of these two conditions is significant in reality.*

My comments below are mostly about Figures: the paper would really benefit from a more consistent representation and discussion of the results and more consistency in the figures with results.

I would also suggest to provide some more informative conclusions, maybe summarized in a bullet point lists.

*Thank you for your comments and suggestions on our manuscript. Comments and suggestions are all valuable and very helpful in revising and improving our paper. We have carefully considered the comments and have made corrections that we hope will meet with your approval.*

**Specific comments**

Line: suggest to change to "across multiple scales"

*Replaced as suggested*

Line 37: I would not say that data scarcity and subsurface heterogeneity are the sole responsible for uncertainty. Depending on the model, we need to add uncertainty on other terms of the water budget.

*Thanks for your comment and suggestion. We acknowledge this limitation in the manuscript in lines 39-42:*

*"There are multiple contributing factors to the uncertainty, including cognitive limitations of models, recharge values, lateral inflows, data scarcity, and subsurface heterogeneity. These factors are responsible for the persistence of uncertainty even after calibration. The latter type of uncertainty is crucial and must be quantified to provide robust decision support in engineering and management practice."*

Line 413: why S1 is a fallback scenario? I wonder if S1 is needed, and if yes I think there is the need to have at least to scenarios with regular well distribution to be able to make a comparison

*Thanks for your comment. We believe that including S1 in the comparison is essential for evaluating the performance of irregular monitoring wells (S0) versus regular monitoring (S1). As mentioned in our paper, the close spacing of some monitoring wells in the irregular networks may present challenges for certain methods due to their high autocorrelation.*

Figure 2: as mentioned earlier: is it a limiting factor having two truths that are so similar and only different for standard deviation?

*Thank you for your comment. Although the two K fields share the same correlation, they have different variances: one with a standard deviation of 2 and the other with a standard deviation of 1. The rationale for utilizing two synthetic truths with varying standard deviations is to assess the efficacy of methods in handling fields with low and high heterogeneity. There are many hyperparameters used for generating the K fields, and it is not realistic to compare the effect of all hyperparameters in one paper. However, we welcome the community to test other hyperparameters in the future.*

Figure 3: I think that the use of a different scale for S2 is misleading, but if the authors believe that it is important I would suggest to add explanation

*Thank you for your suggestion. We have added the following explanation in the caption of the figure:*

*"Please note that S2 exhibits the same standard deviation as synthetic truth 2, while the others exhibit the same standard deviation as synthetic truth 1. Therefore, the scale of S2 should align with that of synthetic truth 2, and the scale of other scenarios should align with that of synthetic truth 1."*

*Besides, we also add the following explanation in the caption of Figure 2:*

*"Please note that to maintain the primary characteristics of the lnK fields given their diverse standard deviations, the scale for these two synthetic truths has been adjusted individually."*

Figure 4: same comment on the scale

*Thank you for your suggestion. We have added the following explanation in the caption of the figure:*

*"Please note that S2 exhibits the same standard deviation of 1 as synthetic truth 2, while the others exhibit a standard deviation of 2 similar to synthetic truth 1. Therefore, to ensure the distinct features of the figures, the scale setting for S2 is adjusted differently from the others."*

Figure 5: same comment: it would be good to add an explanation

*Thank you for your suggestion. We have added the following explanation in the caption of the figure:*

*"Please note that in order to maintain the primary characteristics of the evolution of the RMSE, the scale for these RMSE has been adjusted individually."*

Line 514: it is really hard to see the sharp decrease

*Yes, as indicated in the paper, there is a sharp decrease in RMSE within a brief initial period depicted in the plots. In comparison to the overall extended period, this rapid change during the initial phase is scarcely discernible.*

Figure 6: the legend for S0 says "reference" and for the other scenarios "Synthetic truth". Is this correct? Also, are there some conclusion to be derived from the mean always being above the reference/synthetic truth?

*Thanks for your comment. The term "Synthetic" should also be used, and the figure for S0 has been adjusted. Despite the minor discrepancy, it is still unclear why the mean consistently remains higher than the synthetic truth.*

Figure 9: is the reference solution, the true solution?

*Thank you so much for your comment. The reference solutions are the best possible solutions with best estimate and posterior standard deviation (as mentioned in lines 443-444 of the original version of the manuscript). We have now included the following explanation of the reference solutions in the abstract (lines 6-9).*

*"However, in past studies until now, comparisons were made among approximate methods without firm reference solutions. Note that the*

*reference solutions are the best possible solutions with best estimate, and posterior standard deviation and so forth."*

**Technical corrections**

No technical corrections.